# Direct measurement of warm Atlantic Intermediate Water close to the grounding line of Nioghalvfjerdsfjorden (79°N) Glacier, Northeast Greenland.

Michael J. Bentley[1], James A. Smith[2], Stewart S.R. Jamieson[1], Margaret R. Lindeman[3], Brice R. Rea[4], Angelika Humbert[5,6], Timothy P. Lane[7], Christopher M. Darvill[8], Jeremy M. Lloyd[1], Fiamma Straneo[3], Veit Helm[5], and David H. Roberts[1].

[1]Department of Geography, South Rd, Durham University, Durham, DH1 3LE, UK.

[2]British Antarctic Survey, High Cross, Madingley Rd, Cambridge, CB3 0ET, UK

[3]Scripps Institution of Oceanography, University of California, San Diego, CA, USA

[4]Geography and Environment, School of Geosciences, University of Aberdeen, Elphinstone Road, Aberdeen, AB24 3UF, UK

[5]Alfred-Wegener-Institut, Helmholtz Zentrum für Polar- und Meeresforschung, Bremerhaven, Germany

[6]Faculty of Geosciences, University of Bremen, Bremen, Germany

[7]School of Biological and Environmental Sciences, Liverpool John Moores University, Liverpool, L3 3AF, UK

[8]Department of Geography, The University of Manchester, Oxford Road, Manchester, M13 9PL, UK

Correspondence to: Michael J. Bentley (m.j.bentley@durham.ac.uk)

**Abstract.** The Northeast Greenland Ice Stream has recently seen significant change to its floating margins, and has been identified as vulnerable to future climate warming. Inflow of warm Atlantic Intermediate Water (AIW) from the continental shelf has been observed in the vicinity of the Nioghalvfjerdsfjorden (79°N) Glacier calving front, but AIW penetration deep into the ice shelf cavity has not been observed directly. Here, we report temperature and salinity measurements from profiles in an epishelf lake, which provide the first direct evidence of AIW proximal to the grounding line of 79°N Glacier, over 50 km from the calving front. We also report evidence for partial un-grounding of the margin of 79°N Glacier taking place at the western end of the epishelf lake. Comparison of our measurements to those close to the calving front shows that AIW transits the cavity to reach the grounding line within a few months. The observations provide support for modelling studies that infer AIW-driven basal melt proximal to the grounding line and demonstrate that offshore oceanographic changes can be rapidly transmitted throughout the sub-ice shelf cavity, with implications for near-future stability of the ice stream.

## 1 Background and rationale

The Northeast Greenland Ice Stream (NEGIS) is the largest ice stream in Greenland and the main artery for ice discharge from the northeast sector of the ice sheet to the Fram Strait, draining an area of 200,000 km$^2$ or 12% of the ice sheet (Mouginot et al., 2015). Unlike many other sectors of the Greenland Ice Sheet, NEGIS and the ice shelves that extend from its margin exhibited little response to atmospheric and oceanic warming over the decades immediately prior to the mid-2000s (Khan et al., 2014; Mouginot et al., 2015; Mayer et al., 2018; Lindeman et al., 2020; An et al., 2021). However, recent ice shelf loss and rapid grounding-line retreat post-2010 suggest that this sector of the Greenland Ice Sheet, and NEGIS, are starting to respond to atmospheric/ocean change (Khan et al., 2014). Model projections suggest that ocean warming around Greenland will be double the global mean ocean warming by 2100 (Yin et al., 2011) and air temperature is likely to increase significantly in northeast Greenland (AMAP, 2011; Hanna et al.,

2020). This means that the future evolution of the NEGIS catchment is important, not only for understanding changing dynamics in this sector of the ice sheet but also for predicting future sea level rise as the catchment holds a volume of ice equivalent to 1.1 m of sea level rise (Mouginot et al., 2015; An et al., 2021).

NEGIS extends approximately 600 km from the central ice divide of the ice sheet to the coast, and the ice stream branches into three outlet glaciers (Nioghalvfjerdsfjorden Glacier (79°N Glacier), Zachariae Isstrøm (ZI) and Storstrømmen Glacier (SG); Figure 1) as it approaches the ice sheet margin. The 79°N Glacier grounding line is ~600 m below sea-level whilst upstream of the grounding line the basin floor deepens to ~1000 mbsl (Bamber et al., 2013). 79°N Glacier extends beyond the grounding line in an ice shelf ~80 km long and which thins rapidly from ~700 m at the grounding line to <300m only 10 km further downstream (Fig. 2). Seismic measurements show a deep ocean cavity beneath the ice shelf that extends down to a maximum of 900 m below sea level and a seafloor that rises eastwards towards the calving front, where a sill culminates in several shallow bedrock highs (Fig. 2), which are pinning the ice shelf. (Mayer et al., 2000; Morlighem et al., 2017). Khan et al. (2014) report 25 years of relative stability of both 79°N Glacier and ZI prior to the mid-2000s after which there have been sporadic periods of dynamic thinning and flow acceleration. The ZI ice shelf in particular has undergone rapid retreat (~40 km), and post 2010 it disintegrated completely leaving a grounded tidewater margin which is now retreating along a retrograde slope. Khan et al. (2014) link these changes to increased air temperatures and sea ice loss, with several studies also implicating ocean warming (Mouginot et al., 2015; An et al., 2021) which may have significantly increased submarine melt rates of the ice shelves and the grounded tidewater margins. Humbert et al (submitted) also identify a recent shift in calving style at the southern part of the eastern calving front of the 79°N Ice Shelf;  where normal tongue-type calving has changed to crack evolution, initiated at frontal ice rises and which has progressed >5–7 km further upstream.

A number of studies have highlighted the likely influence of incursions of warm Atlantic Intermediate Water (AIW) on 79°N Glacier and the fronting ice shelf. AIW is a warm, saline water mass of Atlantic origin and which forms from recirculating waters in Fram Strait (Schaffer et al., 2020). Its temperature exceeds 1°C and it underlies cold Polar Water on the East Greenland continental shelf. Schaffer et al. (2017) showed that AIW crosses the continental shelf at depths >150-200 m via Norske Trough and along a continuous deep channel (> 370 mbsl) that extends to the main (southern) calving front of 79°N ice shelf (Schaeffer et al., 2017). This allows AIW to potentially reach the grounding line of 79°N Glacier. Based on the rapid decrease in thickness of the ice shelf by 330 m within 5 km of the 79°N Glacier grounding line, Mayer et al. (2000) estimated very high basal melt rates of 40 m a$^{-1}$ close to the grounding line and suggested these may be due to the presence of AIW. Wilson et al. (2017) used DEMs from optical imagery to estimate maximum grounding line melt rates of 50-60 m a$^{-1}$ reducing to 15 m a$^{-1}$ 15 km downstream The exchange flow across the 79°N Glacier fjord front is primarily controlled by variability in AIW layer thickness, circulation on the continental shelf and bathymetry (Schaffer et al., 2020; von Albedyll et al., 2021). Three gateways have been identified at the main calving front where AIW enters or leaves the sub-ice shelf cavity (Fig. 1a) (Schaffer et al., 2020; von Albedyll et al., 2021). A sill blocks other parts of the southern entrance (Schaffer et al., 2020; von Albedyll et al., 2021) and AIW is unable to enter at the northern calving front in Dijmphna Sund because of a shallow sill (170 mbsl)  that acts to block the deeper inflow (Wilson and Straneo, 2015). There is only one measurement of AIW in the cavity, where it has been detected by an Ice Tethered Mooring (ITM) installed beneath a rift in the 79°N ice shelf, located ~10 km behind the calving front, approximately 13 km north and 18 km west of the primary AIW inflow location (Fig. 1) (Wilson and Straneo, 2015; Lindeman et al., 2020). The ITM from below this rift reveals that AIW is present in the cavity year-round, with maximum temperatures reaching ~1.5°C at 500 m depth in July 2017 (Lindeman et al., 2020). Details of the circulation

of AIW in the cavity remain speculative, as incursion of the AIW into the cavity has not been demonstrated beyond the rift site, but Lindeman et al. (202) suggest that AIW is present down to the seafloor at 720 m depth, immediately below their ITM. Von Albedyll et al. (2021) suggested that a Coriolis-driven circulation would result in the deflection of the main meltwater-enriched outflow to the southernmost part of the cavity. However, data from moorings actually show that while there is some outflow through two of the exits along the southern front, the strongest outflow takes place through Dijmphna Sund, across the northern calving front (Fig. 1) (Lindeman et al., 2020; von Albedyll et al., 2021). The implication is that the sub-ice shelf morphology and/or fjord bathymetry are steering the water circulation within the cavity (von Albedyll et al., 2021).

AIW has warmed in recent years in the Fram Strait (Beszczynska-Moller et al., 2012), in the Arctic Ocean (Polyakov et al., 2012) and in the Norske Trough system (Schaffer et al., 2017). In the latter case a warming of 0.5°C was seen between 1979-1999 and 2000-2016. This appears to be reflected within the 79°N Glacier fjord with CTD profiles showing an increasingly warmer, more saline, and a shoaling AIW layer beneath the ice shelf since 1996 (Mayer et al., 2018; Lindeman et al., 2020). For example, Mayer et al. (2018) indicate an increase of 0.5°C at 175 m depth, which would equate to a potential increase in average ice shelf melt rate of ~5 m a$^{-1}$, following Rignot and Jacobs (2002) who suggested that an increase of 0.1°C in sub-ice shelf water temperature corresponds to a change in basal melt of ice shelves of ~1 m a$^{-1}$.

Mayer et al. (2018) used satellite imagery to track the migration of a compressional ridge (which they referred to as 'Midgardsormen ridge'), formed on the ice shelf surface as it flows obliquely across a grounding line (from floating to grounded) along the northern side of the fjord (Fig. 1a). The migration of the ridge can be linked to changes in ice thickness and can be used to construct a time series of the ice shelf thickness evolution (Mayer et al., 2018). Landward migration of the grounding line/Midgardsormen by 2.1 km between 1994 and 2014 shows ice shelf thinning at an average of 5.3 m a$^{-1}$, with a peak rate of 12 m a$^{-1}$ in 2001-02 (Mayer et al., 2018). Mass balance calculations and flow modelling suggest that glacier dynamics or atmospheric warming alone are unable to explain the magnitude of this thinning trend. However, a simple plume model for the ice shelf cavity showed that penetration of warm water to the grounding line could account for the thinning rate derived from the satellite altimetry (Mayer et al., 2018). This is consistent with findings elsewhere in Greenland where the incursion of warm Atlantic waters to outlet glacier margins over the last few decades, as well as increased air temperatures and sea-ice loss, have all been linked to ice margin instability and rapid grounding line retreat (Vieli and Nick, 2011; Straneo and Heimbach, 2013; Khan et al., 2015; Mouginot et al., 2015).

If AIW is circulating throughout the cavity beneath the floating portion of the 79°N Glacier this is important for grounding line and ice shelf stability (An et al., 2021). Moreover changes in its AIW thermohaline properties may promote further instability. However, future change depends on cavity geometry and the delivery of AIW to the sub-ice shelf cavity and grounding line, and the degree of thermodynamic interactions with the ice shelf base (Wilson and Straneo, 2015; Schaffer et al., 2017; Lindeman et al., 2020; Schaffer et al., 2020; An et al., 2021; von Albedyll et al., 2021). Given the clear importance of AIW in controlling the dynamics and stability of 79°N Glacier it is crucial to understand if it reaches the grounding line, and to characterise 79°N Glacier changes (and any links to AIW), across multiple timescales (Lindeman et al., 2020; von Albedyll et al., 2021). Here we report the first empirical evidence for AIW proximal to the grounding line of the 79°N Glacier based on CTD measurements in Blåsø, an epishelf lake, located >50 km upstream from the calving front (Fig. 1). These are combined with tidal observations, hydrostatic

analysis of the ice shelf, Interferometric Synthetic Aperture Radar (InSAR) and airborne ice-penetrating radar data in order to characterise Blåsø and the contemporary conditions around the grounding line of the ice shelf.

**2 Study Area**

Epishelf lakes occur between ice-free land areas and ice masses floating on the ocean. Where a source of freshwater feeds into the lake, a salinity-driven stratification forms with the more saline marine layer capped by a freshwater layer. The depth of the transition between marine and brackish/fresh water is controlled by the minimum draught of the floating ice (Hattersley-Smith, 1973; Gibson and Andersen, 2002). Epishelf lakes are tidal and can contain mixed biological assemblages of marine, brackish and freshwater organisms. They are well known in polar regions and are a
particularly rich source of information on the adjacent floating ice margins. For example, epishelf lakes have been used to infer past glaciological change such as ice shelf collapse and thickness changes of the floating ice in West Antarctica (Bentley et al., 2005; Smith et al., 2006), East Antarctica (Wagner et al., 2004; Gibson and Andersen, 2002) and the Canadian Arctic (Mueller et al., 2003; Antoniades et al., 2011; Hamilton et al., 2017). Past ice shelf collapses are recorded by changes in the lake sediment being deposited such as loss of freshwater organisms and input of far-
travelled ice-rafted debris (Bentley et al., 2005).

Blåsø is an epishelf lake located adjacent to the northern margin of the floating portion of 79°N Glacier. The lake is broadly triangular in planform with the two southern apexes of Blåsø in contact with two calving margins (west and east), separated by ~10 km (Fig 1). The northern apex of the lake receives freshwater from a glacier-fed river draining from the northwest, which forms a large prograding delta. The eastern and western parts of the lake receive supraglacial
meltwater and any submarine melt from the calving fronts. In late summer 2017 there was continuous lake ice in the parts of the lake within ~1 km of the eastern calving front. Icebergs and floes of lake ice were present in the lake in July-August 2017.

The shoreline of Blåsø has been used to reconstruct past glacier behaviour, with the remains of marine organisms providing clear evidence for retreat of the floating portion of 79°N Glacier and marine inundation of the basin during
the Holocene Thermal Maximum (8.0-5.0 ka BP; Bennike and Weidick, 2001). Lake core evidence has also been recently used to constrain ice shelf retreat, followed by re-advance and epishelf lake reformation at *c.* 4.4 ka BP (Smith et al., in press).

Multiple approximations for the location of the 79°N Glacier grounding line have been produced over recent years (Morlighem et al., 2017; ESA Greenland IceSheet CCI, 2017, Mayer et al., 2018) and they are for the most part in
general agreement, with the exception of the western calving margin in Blåsø (Fig 1a). Morlighem et al. (2017) (Fig. 1a) and An et al. (2021) suggest that the grounding line has retreated beyond the western margin of Blåsø, and that the ice entering the lake is either floating, or very close to it. The other studies indicate that the western end of Blåsø is located somewhere between 1 km and 7 km from the cross-fjord (NW-SE oriented) grounding line with the ice reaching flotation in water depths between 175 m and 450 m. None of the above studies have identified grounding lines along the
lateral margins of the ice stream with the exception of the Midgardsormen ridge identified by Mayer et al. (2018) which lies between the western and eastern calving margins of Blåsø.

**3 Methods**

We mapped the bathymetry of the lake, measured tidal fluctuations, and measured multiple CTD profiles in different parts of the lake. This formed part of a wider programme (Lane et al., 2023; Smith et al., in press; Lloyd et al., submitted) to characterise and sample water and sediments in Blåsø, and ultimately to better understand past and ongoing changes in the 79°N Glacier.

Fieldwork was undertaken at Blåsø between 19th July and 11th August 2017, overlapping with the end of the period of data collection of the ITM reported by Lindeman et al. (2020). We conducted a CHIRP sonar sub-bottom survey to construct a bathymetric map of the lake. The survey was conducted using a SyQuest Bath2010PC sub-bottom profiling unit coupled to a 10 KHz SyQuest P02590-1 transducer. Depth differences at crossovers of sonar survey tracks average 1.6 m but range between -3 and 4.7 m. These differences are likely due to a combination of tides, slight differences in GPS locational accuracy on sloping surfaces, and instrument uncertainty.

CTD profiles were measured with a *Valeport* MiniCTD rated to 500 m with a pressure-balanced conductivity cell, 50 bar strain gauge pressure sensor, and a Platinum Resistance Thermometer sensor. We used the *Datalog X2* software package to record and manipulate CTD data. Manufacturer-cited accuracy was ±0.01 mS/cm, ±0.01°C, and ±0.05% for pressure. These correspond to accuracy of approximately 0.02 g kg$^{-1}$ in absolute salinity, 0.01 °C for Conservative Temperature, and a depth (at our maximum measured depths) of < ±10 cm. The CTD instrument was factory-calibrated two months before the measurements reported here. The CTD was deployed using a hand-spooled winch from a small boat at eight sites across the lake, with the objective of characterising water conditions at both calving fronts and in the three lake basins identified by the CHIRP survey (Fig 1; Table 1). CTD sample rate and the speed of spooling corresponded to a measurement approximately every 10-30 cm. The CTD profiles were sampled between 31st July and 10th August 2017 and during this period there was persistent lake ice which prevented CTD measurements close to the eastern calving front. In contrast, most of the lake ice in the western basin had dispersed and melted by early August, allowing access to the western calving margin. Temperature and salinity observations are reported as Conservative Temperature and Absolute Salinity (McDougall & Barker, 2011).

Over several weeks a *VanWalt* LevelSCOUT water pressure transducer was used to measure any tidal variation within the lake. This uses a silicon strain gauge to measure water pressure variations, with a manufacturer-cited accuracy of 0.05% full scale (FS) and a resolution of 0.0034% FS. In the water depths we deployed the transducer this corresponds to an accuracy and precision better than 1 mm. Water pressure variations were corrected for barometric (atmospheric pressure) variations using measurements from a *VanWalt* BaroSCOUT instrument and the *Aqua4Plus* software package. The LevelSCOUT was secured to a stake submerged in the western end of the lake, and the BaroSCOUT was secured to a stake on the lakeshore <50 m away (Fig. 1b). Overnight on 28/29th July 2017 the tidal pressure transducer stake was impacted by windblown lake ice floes. The transducer continued to record tidal information but with anomalously large water depth readings due to the tilted stake. On 2nd August the lake ice had moved and the stake was re-positioned to the vertical at the same site. The tidal measurements are therefore presented as three measurement intervals: Interval 1 (25th July – 29th July), Interval 2 (29th July – 2nd August), and Interval 3 (2nd August– 12th August) (Fig. 2). These intervals do not have a common datum and so are not directly comparable in absolute elevation, but the amplitude and period measurements are comparable across the three intervals. The low tide minima between 10-12 August are not fully recorded in the dataset due to sub-aerial exposure of the re-positioned sensor at the lowest tides.

We used three independent approaches to determine the patterns of floating and grounded ice at the calving margins in Blåsø, namely hydrostatic analysis, satellite-derived Interferometric Synthetic Aperture Radar, and airborne radio echo sounding data.

Hydrostatic analysis used the Arctic DEM (which is constructed from data acquired over multiple years prior to 2018, and is provided at a ground resolution of 2 meters, for full details see Porter et al., 2018) for the ice surface and BedMachine v3 (which is constructed from multiple ice thickness datasets collected between 1993-2016 and is provided at a ground resolution of 150 m, for full details see Morlighem et al., 2017) for the sub-shelf bathymetry and subglacial topography. We used our CHIRP sonar survey for lake bathymetry, and the MEaSUREs Greenland Ice Sheet Velocity Map (Joughin et al., 2016; 2017) for surface velocites. Elevations from the Arctic DEM were corrected to orthometric heights using the EIGEN-6C4 geoid from BedMachine v3 (Morlighem et al., 2017). An approximation for the ice shelf thickness, where it is in hydrostatic equilibrium i.e. the ice surface elevation is a direct measure of the local ice thickness, was made using an ice density of 917 kg m$^{-3}$ and sub-shelf water density of 1023 kg m$^{-3}$ (following Schaffer et al., 2016). This was also used to approximate the location of grounded, partially grounded, and floating ice.

We applied Interferometric Synthetic Aperture Radar (InSAR) on Sentinel-1 TOPS SAR data captured on 3, 9, and 15 April 2017. Interferograms are formed from interferometric wide-swath single-look complex data acquired at times t1 and t2 separated by a temporal baseline of 6 days. Assuming constant horizontal ice flow within a 6-day time period we subtracted another interferogram with data acquired at times t2 and t3 to isolate vertical displacements due to ocean tides. Areas of scatter (low coherence) on the interferograms represent areas of ice shelf bending close to grounded ice. Full processing details can be found in Christmann et al. (2021).

We used airborne ice-penetrating radar data to examine the relationship between grounded and floating ice at the eastern calving margin in Blåsø. The data were from the Alfred Wegener Institute (AWI) multi-frequency ultra-wideband (UWB) radar. This is a modified version of the Multichannel Coherent Radar Depth Sounder (MCoRDS), that is mounted on the *Polar 6* Basler BT-67 aircraft. A laserscanner continuously records the elevation of the ice surface in a swath beneath the aircraft. Acquisition parameters can be found in Franke et al (2020) and data were collected on 14[th] April 2018, approximately 8 months after the field measurements undertaken in Blåsø. The geographic precision of the flight trajectory and, thus, both the radar and laserscanner data is around 0.05 m. Uncertainty of the GNSS altitude and, thus, laserscanner elevation is usually within 0.1 m. The laserscanner is calibrated using runway passes and runway crossing. The uncertainty of picking the bed or ice/water interface ice can be influenced by side returns, hyperbolic reflections from basal crevasses plus uncertainties in propagation velocity but are usually well within 10 m.

## 4 Results

Blåsø has an area of 60.7 km$^2$ and 53.5 km of shoreline. The bathymetric survey of the lake reveals a western basin (maximum depth ~135 m) in front of the west calving front, a central basin (~90 m deep) and an eastern basin (maximum measured depth ~212 m) close to the east calving front (Figure 1b). The central and eastern basins are separated by a broad sill at 57 m depth in the central part of the lake. The western basin is separated from the central basin by a sill at 21 m depth (Fig. 1b) that is consistent with the continuation of a subaerial moraine visible on the western shore. The northwestern parts of the lake are very shallow (<20 m) and are occupied by the lower reaches of a large delta. The central basin has a flat floor in which the CHIRP identified stratified sediment.

Figure 1b shows the location of the *VanWalt* LevelSCOUT water pressure transducer. Despite the complications due to the transducer being impacted by an ice floe, the data clearly demonstrate that the lake experiences a semi-diurnal tidal cycle (Fig. 3). The maximum tidal range of ~1.2 m was only fully recorded during Interval 1. The second half of

Interval 2 and the early part of Interval 3 record have the lowest tidal range of ~0.4 m. Spring and neap tides were captured within the record.

We obtained eight CTD profiles in Blåsø (Fig. 1, Fig. 4, Table 1). These ranged from 15 m depth around shallow margins of the central basin to 192 m depth in the eastern basin. The deepest profiles were retrieved close to the eastern end of the lake where the CHIRP bathymetry shows the lake deepens significantly towards the calving front. In the western basin CTD profiles were obtained at a maximum depth of 136 m, immediately adjacent to the calving front.

The profiles show a stratified water column, marked by two prominent pycnoclines where temperature and salinity
show abrupt changes with depth. Our measurements show that Blåsø corresponds to a Type I epishelf lake (Gibson and Anderson, 2002) with stratified freshwater overlying marine water. We identify four primary water masses within the lake. By water mass we mean regions of the lake with similar temperature and salinity properties but contrasting to other water found above/below or laterally (Figures 4, 5). The surface water mass is a shallow freshwater cap (identified here as water mass 1; salinity <1 to 2 g kg$^{-1}$), which is approximately 20 m thick. The surface is cooler in the eastern
basin (2.3-2.7°C), where persistent lake ice also trapped a large volume of icebergs, than in the western basin (3.1-4.5°C) where little lake ice remained into August. Surface temperature also varied with distance from the calving front within the western basin: the profiles CTD3 and CTD4 show warmer surface temperatures than profiles CTD6 and CTD7 that were taken within 100 m of the calving front (Fig. 1b, Fig. 4).

Below water mass 1 is an intermediate brackish layer occupying a depth range from 20 m to 145 m. Properties of this
layer vary between the west and east basins. In the eastern basin, salinities are in the range of 5 to 7 g kg$^{-1}$ and temperatures 0.5 to 0.8°C and this is identified as water mass 2. In the western basin, the layer is more saline (11-13 g kg$^{-1}$) and colder (-0.1 to -0.5°C) and is identified as water mass 3.

Present only in the eastern basin and below 145 m is a seawater layer (water mass 4) which extends to the deepest part of the lake sampled (192 m). Water mass 4 is relatively warm and salty, with temperatures between 0.1 and 0.7°C and
260 salinity 34.4 to 34.7 g kg$^{-1}$. The lower pycnocline shows an offset of ~5 m in the measured depth of the pycnocline boundary between CTD5 and CTD8 in the eastern basin, with CTD8 showing a slightly shallower pycnocline than three days earlier in CTD5.

During bathymetric and CTD surveys, we observed dead and dying (on their sides, rapid gasping) fish floating at the surface of the lake on several occasions. These were only seen in the western basin, close to the calving front, and
265 above the deepest point. The fish were Arctic cod (*Arctogadus glacialis*), a marine fish that is known to live in close proximity to ice (Froese and Pauly, 2021).

Our three approaches to determining the distribution of floating and grounded ice yield slightly different results, with both the interferometry and radio-echo sounding data suggesting that much of the eastern calving margin is floating, but hydrostatic analysis suggesting a narrow grounded zone exists close to the calving front (Figs. 6, 7, 8)

Our new hydrostatically-based estimate for the grounding line has been made across the width of 79°N Glacier and at both of the calving margins which enter Blåsø. Both of these have been tied, as far as is possible, to the CHIRP bathymetry surveys (Fig. 6). The grounding line across most of the fjord is in general agreement with the previous work (Fig. 1a). The location of the grounding line around the western entrance into Blåsø is more complex to interpret. Between 1994 and 2014 Midgardsomen migrated towards the ice shelf margin (~2 km), and moved further down stream
from the western calving margin of Blåsø (Mayer et al., 2018). The ice sheet/shelf surface slope transition, observed in the Arctic DEM, appears to be located up-flow from the western entry to Blåsø. These observations support the

interpretation of Morlighem et al. (2017) (Fig. 1a) and An et al. (2021) that the across-fjord portion of the grounding line is located up-flow of Blåsø (Fig. 6d) and the ice flowing into the western entry of Blåsø is floating. However, the ice surface elevation, taken from the Arctic DEM, suggests that the ice is mostly grounded across the full width of the western margin of Blåsø i.e. the ice is higher than it should be if the ice were free floating (Fig. 6d). The data are equivocal so two possible locations for the grounding line are shown on Figure 6 with the most likely scenario being that the ice is only partially grounded in the area between the two dashed lines (Fig. 6) as indicated in Figure 6d.

The hydrostatic analysis suggests that much of the ice draining into the eastern end of Blåsø appears to be grounded, due to the ice surface elevation being greater than that required for the ice to be floating and in hydrostatic equilibrium. This is supported by the presence of the Midgardsormen ridge (Mayer et al., 2018), which runs along the northwestern margin of the ice shelf and across the eastern entry into Blåsø (Fig. 6e). There is, however, an area of floating ice that extends west from Midgardsomen and reaches to within 300 m of the eastern calving front in Blåsø (Fig. 6c). Figure 6 shows profiles across the eastern calving front with one oriented along the deepest centre-line of the eastern basin of Blåsø (Fig 6e) and a second aligned through the area of floating ice closest to the calving front and which terminates on the shallower (~50 m depth) southern flank of the eastern basin (Fig 6f).

We are able to further resolve the extent of grounded and floating ice at the two calving fronts in Blåsø using interferometric analysis (Fig 7). Double differential interferograms from data captured on 3, 9 and 15 April 2017 help demonstrate the extent of grounded and floating ice. At the western calving margin the band of fringes showing grounded ice extends across the calving front. At its narrowest this grounded ice only extends a few 100 m behind the calving front (Fig 7a). A small ice rise (grounded ice within the floating ice shelf) is evident ~3 km to the south of the calving front. A double differential interferogram of the eastern calving front provides a more detailed view than the hydrostatic analysis and shows that unlike the western margin the calving front is not grounded (Fig 7b) over most of its area. A small ice rise and narrow zone of possible grounding along the Midgardsormen is evident 1-2 km behind the calving front.

Our third approach, using ice penetrating radar data, shows that the eastern margin is actually floating on both sides of Midgardsormen (Fig. 8) close to the eastern calving margin. A prominent reflector shows the ice shelf base with a draught in the lake of approximately 150 m (Fig 8a) immediately behind the eastern calving front. The laser scanner data show the ice surface to be higher at the location where interferometry also suggests there is an ice rise (Fig 8b).

**5 Discussion**

Our CTD observations demonstrate four water masses in the lake. A topmost water mass comprises a fresh layer likely derived from runoff or surface melt of floating ice. This layer is confined to the top ~20m which also happens to be the depth of the sill that confines the western basin, so it can spread between the different basins unimpeded. The properties of this water mass are quite similar within all basins although the temperature and salinity of the topmost few metres varies according to freshwater input, lake ice melting and proximity to calving fronts.

An intermediate layer comprises two brackish water masses whose properties are distinct between the eastern (water mass 2) and western (water mass 3) basins. This is consistent with the presence of the 20 m sill which keeps the east and west basin waters within this layer separate. Likely these water masses are affected by the iceshelf/icebergs that confine them. Their origins are unclear but wintertime vertical mixing of deeper (saltier) water masses could play a role. The 4[th] water mass is found in the parts of the eastern basin deeper than 145 m.

Recent studies have suggested that thinning of the 79°N Glacier ice shelf near the grounding line over the past two decades has been driven primarily by warming of ocean waters within the cavity (Mouginot et al., 2015; Mayer et al., 2018; Schaffer et al., 2020; Lindeman et al., 2020). However, this warming has been inferred from limited observations in the vicinity of the calving front, over 70 km away from the grounding line, due to the difficulty of accessing the sub-shelf cavity. The data presented here demonstrate that the seawater properties observed between 160 m and 193 m in the eastern basin of Blåsø lie on a meltwater mixing line (Gade, 1979) between the properties of glacial melt-modified AIW recorded by the ITM deployed beneath the rift in the ice shelf near the calving margin in July 2017 at 150 m and 250 m (Figure 4b; Toole et al., 2016). This provides the first direct evidence that warm AIW is circulating deep within the sub-ice shelf cavity and reaches proximal to the grounding line of 79°N Glacier. The CTD observations made in Blåsø, some 50 km upstream of the calving front appear to reflect similar conditions to those recorded by the ITM proximal to the calving front.

Lindeman et al. (2020) show that trends in the AIW inflow properties at the ITM, observed at 500 m depth, are reflected in the exported melt-modified waters at 250 m with a lag of approximately 180 days, i.e., outflow properties at 250 m in July 2017 fall along a melt mixing line relative to inflow properties at 500 m depth in January 2017. In the eastern basin of Blasø in August 2017 (CTD 8), T-S properties below 175 m fall along the same mixing line (Figure 5b). This indicates that variability in AIW inflow properties, observed near the calving front, propagates into the sub-ice shelf cavity, to within 20-30 km of (and probably all the way to) the grounding line (Schaffer et al. 2020, Fig. 5), in less than 180 days. Flow of melt-modified AIW from the continental shelf to the grounding line is consistent with the conclusions of Wilson and Straneo (2015) and Schaffer et al. (2020), who calculated the residence times for water in the ice shelf cavity as 110-320 days and 162 days respectively. The data presented here lie within this range and further constrain this transmission. The modification by sub-ice shelf melt is considerable: inflowing AIW at 500 m depth close to the calving front is 1.1 – 1.5°C and saltier than 34.9 – 35.0 g kg$^{-1}$ but by the time it reaches Blaso it is 0.1-0.7°C and 34.4-34.75 g kg$^{-1}$.

The water pressure transducer data demonstrate clearly that Blåsø is tidal, confirming connection to the ocean. The tides recorded in Blåsø are consistent with the frequency and range of tidal measurements made by Reeh et al (2000) in open water close to Bloch Nunatakker (referred to as "Syge Moster" in Reeh et al.) at the calving front of the floating ice shelf of 79°N Glacier in 1997 (Reeh et al., 2000) (Fig. 1a). The 1997 measurements were recorded only for a few days but themselves are similar to the tidal measurements at the Danmarkshavn tide gauge (76° 46' N, 18° 46' W; Fig. 1a) with a lead of approximately 50 minutes at Bloch Nunatakker. Reeh et al. (2000) also measured tidal motion of the ice shelf itself using GPS receivers along a cross-glacier profile a few kilometres east of Blåsø. The tidal motion of the ice shelf leads the tidal cycle by ~65 minutes and has a tidal amplitude ~75% of that measured at Danmarkshavn (Reeh et al., 2000). The tidal signal recorded in Blåsø shows a lead of ~1 hour and a tidal amplitude ~65% of that recorded at Danmarkshavn (Fig. 3) consistent with the measurements made by Reeh et al. (2000).

CTD data show two pycnoclines in the lake. The 20 m surface layer likely represents a combination of surface heating and the presence of a lake floor moraine which acts as a sill at 21 m depth preventing exchange of deeper water from the western basin into the central basin. The pycnocline at 145 m acts as a proxy for the minimum draught of the floating ice shelf of 79°N Glacier at the eastern margin. In waters shallower than this the freshwater runoff from the catchment, the surface of the ice shelf, and submarine melting is impounded by the calving cliffs. Deeper than this there is tidal exchange of seawater that comes from under the floating ice shelf of 79°N Glacier.

The difference of ~5 m between the depths of the lower pycnocline measured in profiles CTD5 and CTD8 (Fig. 3) cannot be explained by barotropic tidal variation (1.2 m) alone. CTD5 was sampled on 7th August close to low tide, whilst CTD8 was sampled on 10th August, 1 hour after high tide. Other explanations might include one or more of internal waves, tidally-driven flow up and down isopycnals, or that the pycnocline is shoaling or deepening on a seasonal basis. Internal waves can be created where tidal currents drive water parcels, especially, on steep slopes (Munk and Warren, 1981). Our newly-measured bathymetry presented here shows that the lake deepens rapidly in its eastern end with a slope from the central sill towards the eastern calving front. Such slopes, along with a tidal forcing, are conducive to the development of internal waves along boundaries between water layers of different density, as seen in the eastern basin. Internal waves are of significantly larger amplitude than any surface expression, and move more slowly (Sverdrup et al., 1942). Hence, the offset between CTD5 and CTD8 may be the result of an internal wave propagated through the eastern basin. Such waves have been shown to drive high frequency (50 mins) variability in pycnocline depth at the Milne Fiord epishelf lake in the Canadian Arctic but these internal waves have a maximum amplitude of 0.15 m (Hamilton et al., 2017) and so internal waves of 5 m seem unlikely in Blåsø. Seasonal shoaling of the pycnocline, as reported in Milne Fjord epishelf lake, may also have occurred, especially if water is escaping through eroding/melting cracks and channels in the ice shelf base (Bonneau et al., 2021). From the existing data we cannot confidently attribute the difference to one of these origins, but this does not affect our broader conclusions.

BedMachine infers a steep slope on the side of the fjord under the 79°N ice shelf (Fig. 7), but we note some of the limitations of inferring bathymetry beneath floating ice. The ice surface elevation and the Midgardsormen-defined grounding line (Fig. 6, 7b, 7c) indicate that much of the ice flowing into the eastern basin is grounded along the fjord side, but the 145 m-depth halocline is close to the draught of the floating part of the ice shelf, on both sides of where it is grounded at the Midgardsormen (Fig 8a). CTD5 and CTD8 demonstrate a direct connection to the sub-shelf water circulation and data from the *VanWalt* LevelSCOUT water pressure transducer imply that there is tidal exchange of seawater between the lake and the sub-shelf cavity. The interferometric analysis and radar data show this floating ice at the eastern margin clearly (Fig 7b, Fig 8a) and demonstrate an ice shelf draught of approximately 150 m, consistent with the halocline measured at CTD5 and CTD8. The difference in patterns of grounded and floating ice between the interferometric and radar datasets on the one hand and the hydrostatic analysis on the other, suggest that the lower resolution of the hydrostatic analysis may fail to capture smaller-scale detail of the bed or that it is perhaps affected by the areas of local grounding demarcated by the interferometry, and which may allow the ice to be supported at higher elevations, or 'bridged' between grounded areas.

It is also likely that hydrostatic analysis would not pick up areas of local floating ice such as associated with sub-ice channels, or basal crevasses below the resolution of the data used in hydrostatic analysis. The Midgardsormen splits as it bends slightly into the eastern margin of Blåsø, mirroring an embayment in the fjord bathymetry (Fig. 6 inset). This is interpreted to represent the most likely location of the connection between the deeper parts of the lake and the sub-ice shelf environment. In later imagery from 2020 (Fig. 1b), it appears that the Midgardsormen has migrated farther landward as the ice shelf thinned, suggesting that the connection between the eastern basin and the sub-ice shelf cavity may only recently have been created or has enlarged in recent years.

Our observations of brackish water and dying marine fish show that there is also some connection between the ocean waters and this epishelf lake at the western margin. As there is no fully marine water present in the deepest parts of the western basin, this implies that the seawater layer in the sub-shelf cavity lies deeper than the lake bed (136 m at CTD 7). This is consistent with either of the dashed grounding lines shown in Figure 6. However, the salinities of >10 g kg⁻

 The observations of dying marine
fish at the surface of the western basin is further compelling evidence of an ongoing marine connection. We suggest that
marine fish present in the seawater under the ice shelf are transported into the lowermost brackish parts of the western
basin, and float to the surface when their salt balance and buoyancy are disrupted due to osmotic lysis in their cells. The
most likely explanation is that the ice is only partially grounded between the fjord and the western calving front. Spring
tides, internal waves or flow of buoyant sub-ice shelf meltwater could create a mechanism that pumps seawater (and
fish) sporadically beneath the partially grounded ice into the deepest part of the western basin. This part of Blåsø may
correspond to a Type 2 epishelf lake (Gibson and Anderson, 2022) where the boundary between lake water and fully
marine water lies seaward of the calving front and there is an indirect connection from the lake to the marine
environment. The interferometric analysis shows that at its narrowest the zone of grounded ice at the western margin is
only a few 100 m wide and it is possible that local thinning, sub-ice channels or basal crevasses may allow seawater to
enter the lake.

Analysis of the margins and marine connections in both the western and eastern basins, allied to the time series of
marginal ice shelf thinning from Mayer et al (2018) clearly indicate that the ice shelf margin has thinned sufficiently to
begin un-grounding and allow the ingress of marine water into the western part of Blåsø and, to penetrate further into
Blåsø in the eastern part (as the pycnocline shoals with ice shelf thinning). This represents a critical process in the
retreat of a marine-terminating glacier such as 79°N Glacier. Continued thinning would lead to complete ungrounding
and penetration of warmer marine waters further into shallower parts of Blåsø. Eventual ice shelf removal would mark
its transition to a fully marine basin, analogous to its configuration during the early Holocene (Bennike and Weidick,
2001; Smith et al., in press).

## 6 Conclusions

The measurements reported here provide the first observation of melt-modified Atlantic Intermediate Water (AIW)
immediately proximal to the grounding line of Nioghalvfjerdsfjorden (79°N) Glacier. Measurements show that the
water below 145 m depth in the eastern basin of Blåsø has properties that lie on the mixing line of inflowing AIW
measured approximately six months previously in a rift mooring ~50 km away (Lindeman et al., 2020). Our
observations are also consistent with other estimates of circulation time under the ice shelf.

We show that different approaches to determining patterns of floating and grounded ice at the margin of the ice shelf
can yield significant differences, with ice radar and interferometry yielding a more consistent pattern than hydrostatic
analysis, where patterns are likely complicated by local bed and ice shelf base topography, and potential stress bridging.

Recent studies have emphasised the importance of monitoring water mass properties entering the 79°N Glacier sub-ice
shelf cavity and understanding the associated circulation (e.g. Lindeman et al., 2020; Schaffer et al., 2020). This study
shows that melt-modified AIW is present in the eastern basin of Blåsø, at depths >145 m, and therefore must be present
under the adjacent ice shelf. Our observations are from a location where ocean variability and the grounding line
evolution could be relatively easily monitored. Monitoring the intrusion of AIW beneath the ice shelf, including
changes to the water mass properties, may be possible by future installation of oceanographic moorings in Blåsø, ideally
in both the western and eastern basins. Combined with observations of AIW changes outside the ice shelf, this would
allow monitoring of their propagation into the cavity toward the grounding line of the largest ice stream in Greenland,
potentially revealing variability in circulation and susceptibility of ice shelf melt to ocean warming. Such oceanographic
measurements close to grounding lines are normally only possible by sporadic drilling through ice shelves which are

high cost and logistically difficult. CTD monitoring in the western basin would identify the further un-grounding as the margin thins to a critical thickness that will allow the establishment of a permanent connection between the western basin and the sub-ice shelf cavity.

The observation of melt-modified AIW proximal to the grounding line has significant implications for the 79°N Glacier because basal melting is critical to its stability (Mouginot et al., 2015; Schaffer et al., 2017; An et al., 2021). Our observations confirm modelling studies that have assumed circulation of AIW throughout the entirety of the cavity beneath the 79°N Glacier floating ice shelf whilst being mixed with glacial meltwater from basal melting of the ice shelf. Future modelling of 79°N Glacier evolution can use these observations to constrain water mass properties at the grounding line of this glacier which drains 6.28% of the Greenland Ice Sheet (Krieger et al., 2020). Our data on the circulation times within the cavity provide an empirical constraint on how quickly warming of the AIW, that has been observed on the continental shelf, is transmitted to the grounding line of 79°N Glacier (Wilson and Straneo, 2015; Schaffer et al., 2017; Schaffer et al., 2020).

**Data Availability**

CTD data (Bentley et al., 2023a) and pressure transducer data (Bentley et al., 2023b) from Blåsø are available from the UK NERC Polar Data Centre (http://www.bas.ac.uk/data/uk-pdc/).

**Author contributions**

MB, DR, JS, JL conceptualised the epishelf lake measurements. MB, JS, SJ undertook field measurements in Blåsø aided by CD, DR, TL whilst ML, FS were responsible for collection and interpretation of sub-ice shelf mooring data. AH and VH collected and analysed airborne radar and satellite interferometric data. MB analysed pressure transducer data and BR undertook hydrostatic analysis. All authors contributed to data interpretation. MB led writing of the manuscript with DR, BR, and all authors commented on the manuscript.

**Competing interests**

The authors declare that they have no conflict of interest.

**Acknowledgements**

This work was carried out as part of UK NERC Grant, NE/N011228/1 'Greenland in a warmer climate: What controls the advance & retreat of the Northeast Greenland Ice Stream'. We thank the Alfred Wegner Institute, in particular Hicham Rafiq, for their significant logistic support through the iGRIFF project to the work reported here. The airborne data were acquired as part of the campaign RESURV79 conducted by Polar 6. We also thank Jorgen Skafte (Villum Research Station), Nordland Air, Air Greenland, Joint Arctic Command (Station Nord) and Nanu-Travel (in particular Isak and Ooni) for their role in supporting the fieldwork at Blåsø. Naalakkersuisut, Government of Greenland, provided Scientific Survey (VU-00121) and Export (046/2017) licences for this work. We also acknowledge help from the Danmarks Meteorologiske Institut for supplying tidal data, and the support of the Department of Geography, Durham University. Chris Orton (Durham Geography) made Figure 6d-f. Comments from Derek Mueller, Jérémie Bonneau and an anonymous reviewer helped strengthen and clarify the paper.

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

 **Figures and Tables**

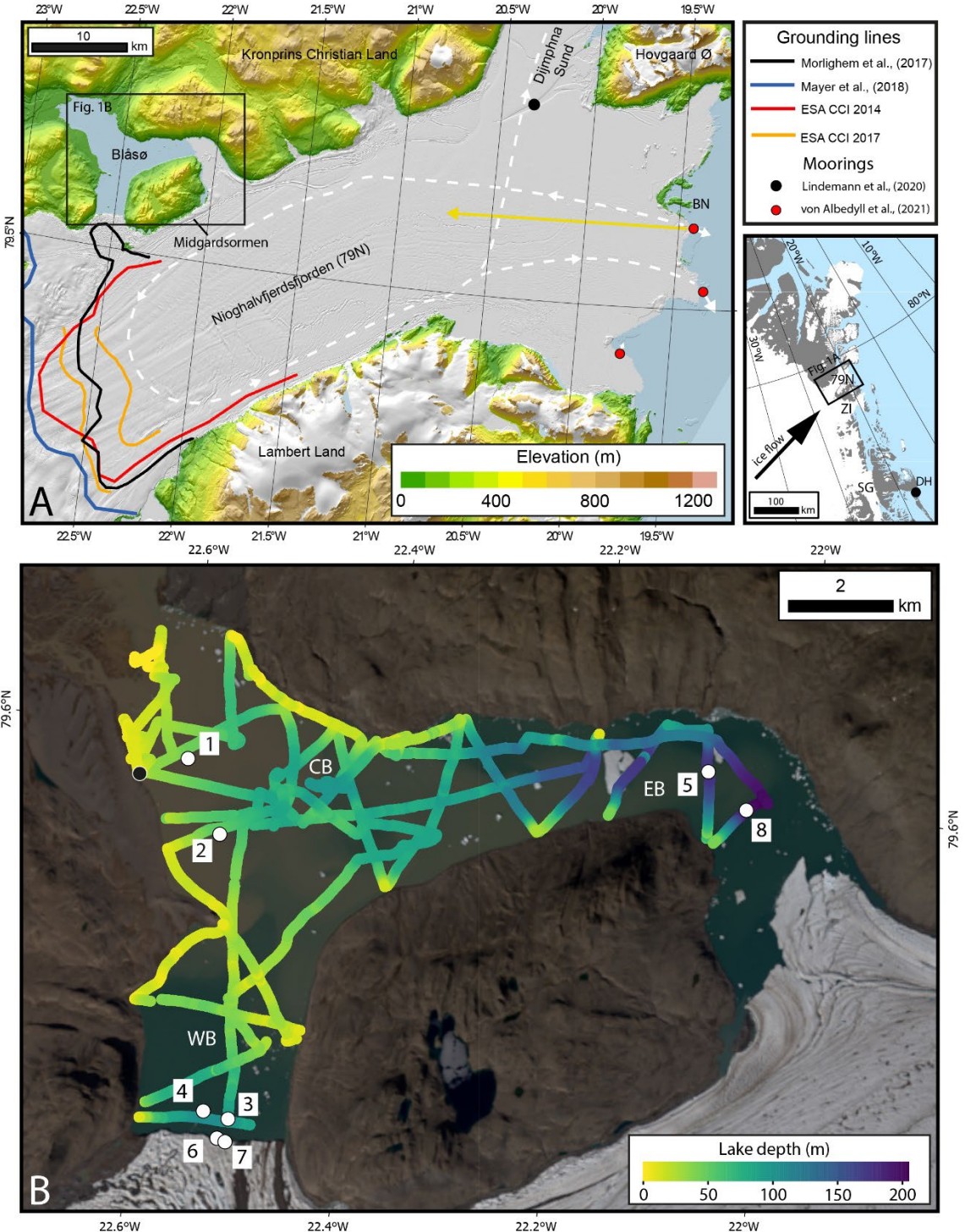

**Figure 1**. Location Map of Blåsø. **(A)** Nioghalvfjerdsfjorden floating ice shelf and the location of Blåsø on its northern side, and the 79°N Glacier grounding line (Mayer et al., 2018; Morlighem et al., 2017; ESA CCI v.1.3) in the context of regional topography from the ArcticDEM (Porter et al., 2018). Black dot marks location of rift mooring of Lindeman et al (2020). BN=Bloch Nunatakker. Red dots show where moorings have measured flow direction in (yellow arrow) and out (white arrows) of the cavity. Arrow lengths are scaled by current speed (von Albedyll et al. 2021). Dashed lines show a potential cavity circulation (von Albedyll et al 2021). Inset shows location of Fig. 1a, ZI = Zachariae Isstrøm, SG = Storstrømmen Glacier, DH=Dansmarkhavn. **(B)** Blåsø bathymetry and CTD locations. Colour ramp along transect lines shows surveyed water depths using CHIRP and demonstrates the presence of a Western basin (WB), central basin

(CB) and Eastern basin (EB) in relation to the western and eastern calving fronts. Locations of numbered CTD sites are shown as white dots and the tide gauge is shown as a black dot. Satellite image is Landsat 8, from 2$^{nd}$ August 2020. Note that during July-August 2017 the eastern end of the lake S and E of the site of CTD8 was occupied by floating lake ice, the edge of which was close to the bathymetric survey line passing through CTD8 and thus the depth of water at the eastern calving margin is not known.

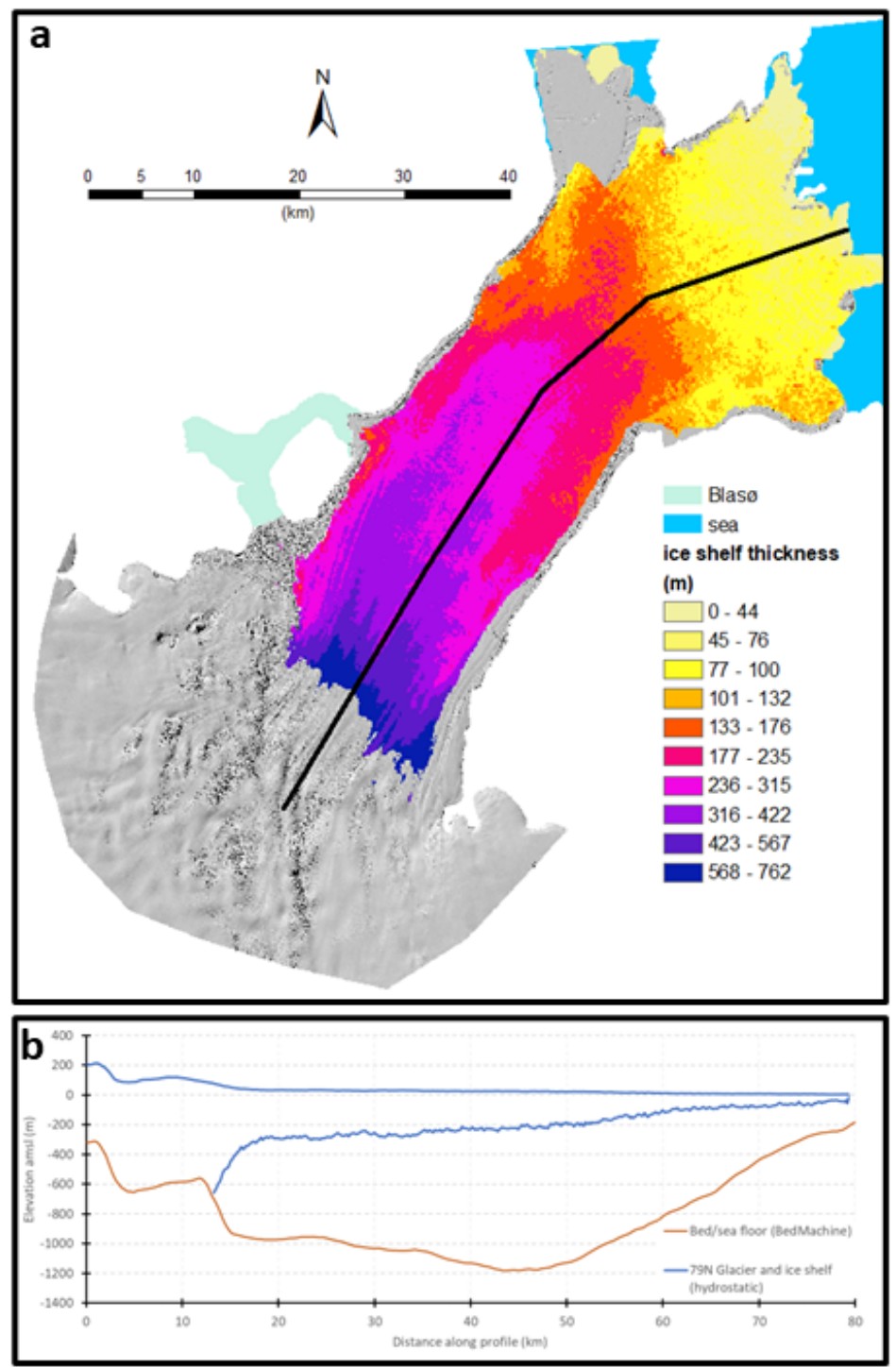

Figure 2. Ice Shelf thickness and cavity morphology. (a). Map of the floating ice shelf thickness, calculated from the ice surface elevation assuming the ice is in hydrostatic equilibrium, overlaid on the ArcticDEM. The ice shelf margins are deemed grounded where the calculated ice shelf draft is greater than the bathymetric depth. B. Long profile of ice shelf thickness and fjord floor along the solid black line in panel (b). Ice surface elevations

and the ice stream bed and fjord bathymetry are all from BedMachine (Morlighem, 2017).

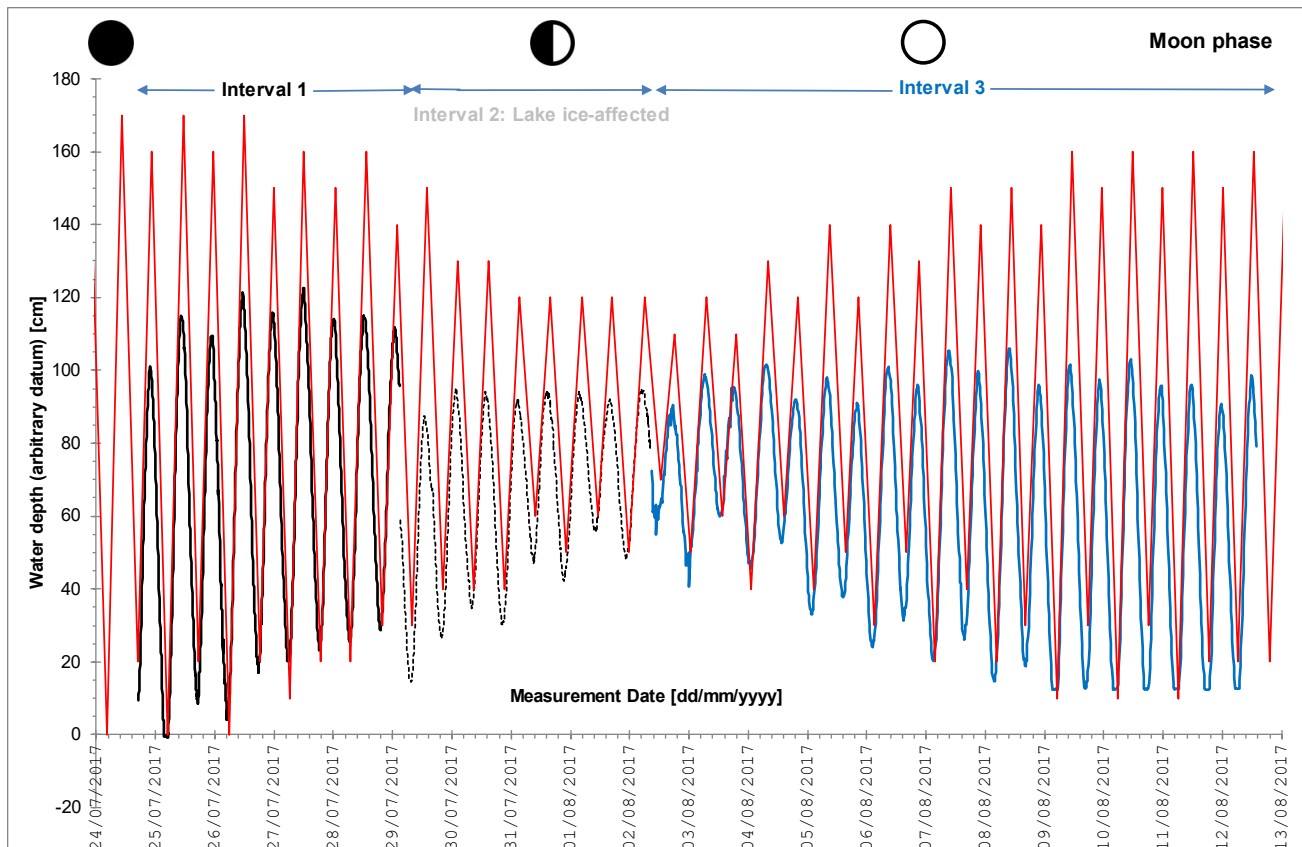

**Figure 3.** Record of water level fluctuations at Blåsø, 25 July 2017 to 12 August 2017. Intervals 1 (black), 2 (black dashed) and 3 (blue) are described in the text but for clarity the intervals have been displaced vertically such that each interval is centred around the average value for that interval. The different intervals therefore have different (arbitrary) datums. Red line shows tidal data for Dansmarkhan tide gauge (Ribergaard. 2017). Moon phases are shown (New moon - 23/7/17, First quarter - 30/7/17, and Full Moon - 7/8/17).  All times are Universal time. The pressure transducer record of water fluctuations demonstrates the semi-diurnal tidal cycle in the Blåsø epishelf lake.

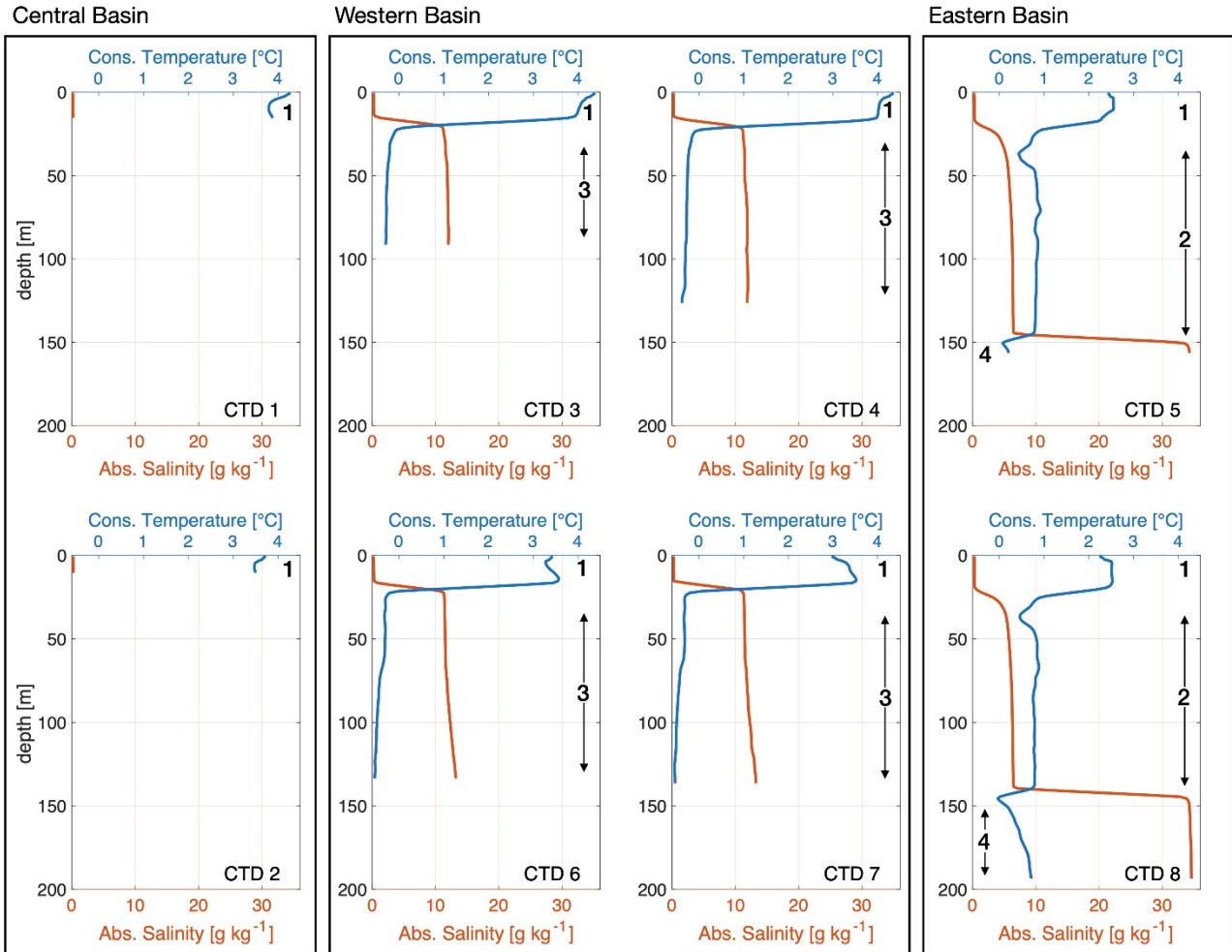

**Figure 4.** Temperature and salinity profiles observed at locations indicated on Figure 1b and in Table 1, arranged by basin. Bold numbers refer to Water Mass numbers discussed in text.

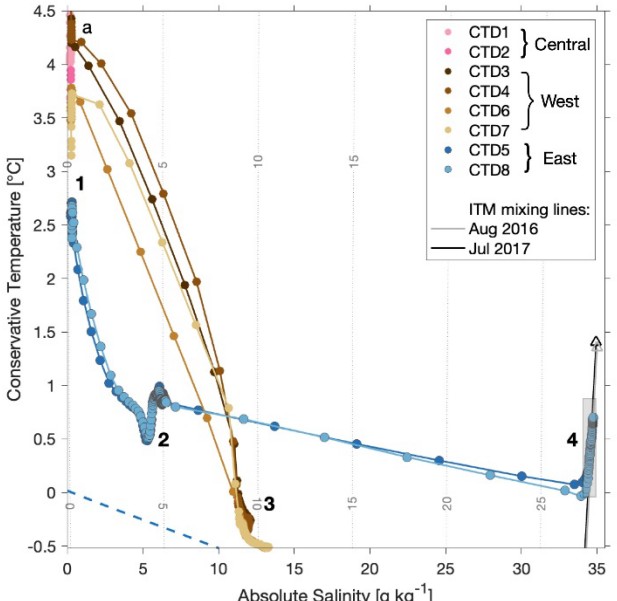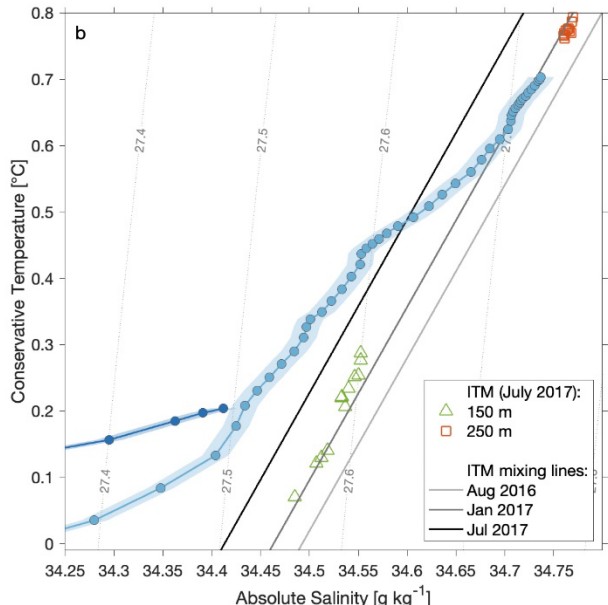

**Figure 5. (a)** Temperature-salinity plot of the profiles shown in Figure 3. Observations from the eastern basin
(CTDs 5 and 8) are in blue. Water masses 1-4, as described in the text, are labelled. The melt mixing lines
correspond to the inflowing AIW properties observed at 500 m depth in the rift by the ITM in August 2016
(open grey triangle) and July 2017 (open black triangle) (Lindeman et al., 2020). The dashed blue line indicates
the surface freezing temperature. Shaded grey box shows extent of Fig 5b.

**(b)** Temperature-salinity plot of data for water mass 4 between 140 and 193 m depth in the eastern basin, only.
The shaded range indicates the instrument precision. Melt mixing lines relative to inflowing AIW properties at
500 m depth at the ITM site are plotted for August 2016 (light grey), January 2017 (dark grey), and July 2017
(black). Daily values of temperature and salinity for out-flowing melt-modified AIW observed in July 2017 by
the ITM are plotted for 150 m (green triangles) and 250 m (red squares) depths.

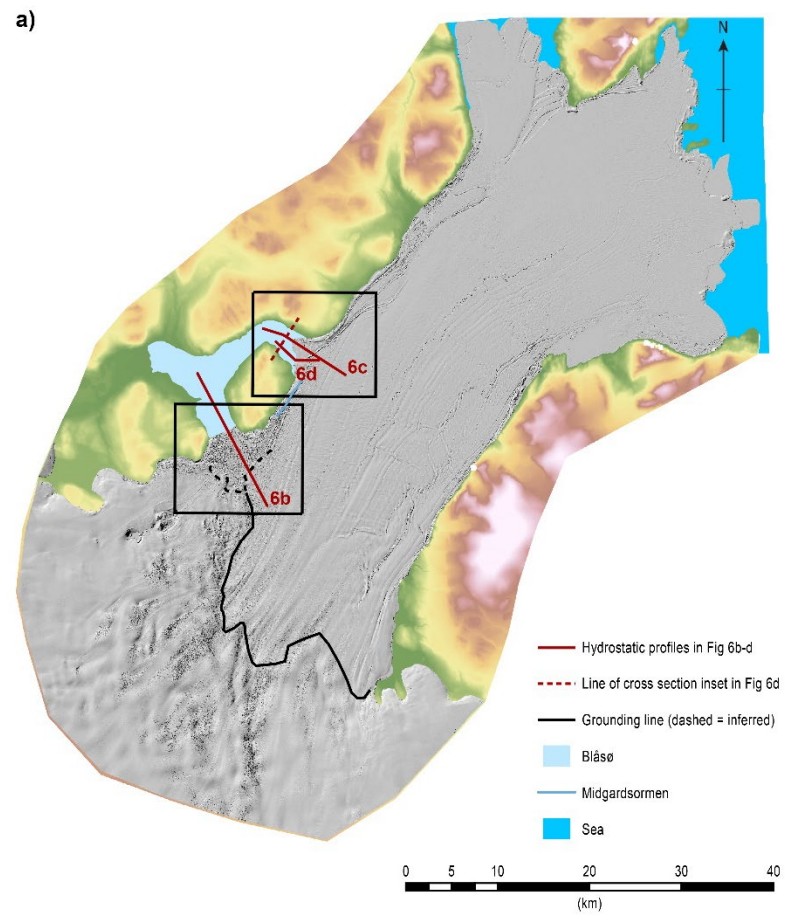

Hydrostatic profiles in Fig 6b-d
Line of cross section inset in Fig 6d
Grounding line (dashed = inferred)
Blåsø
Midgardsormen
Sea

0  5  10      20      30      40
(km)

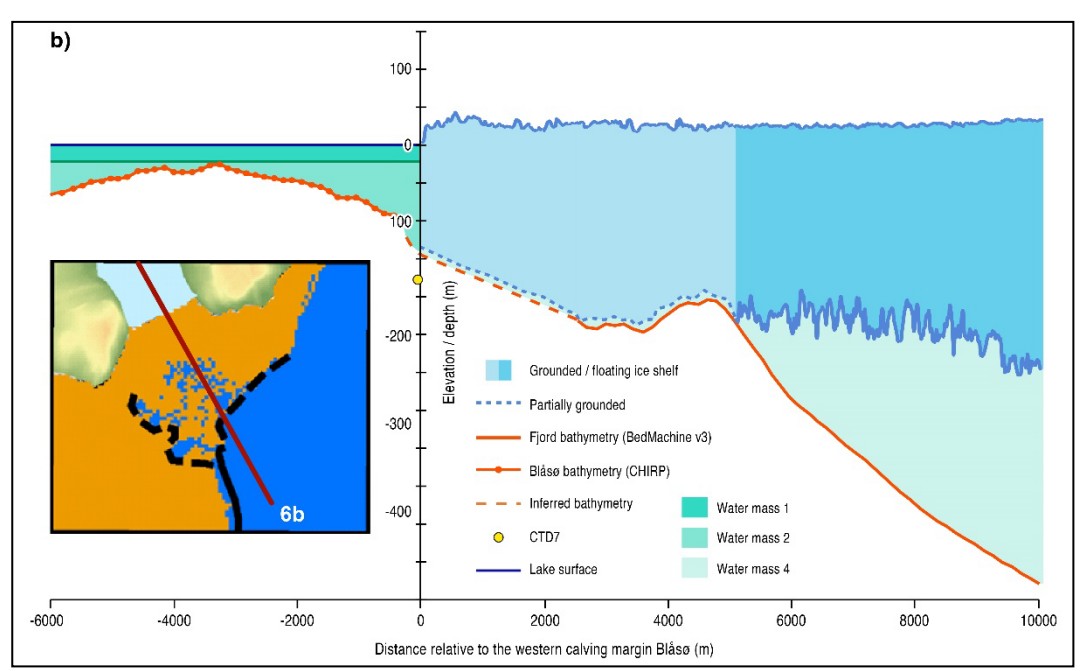

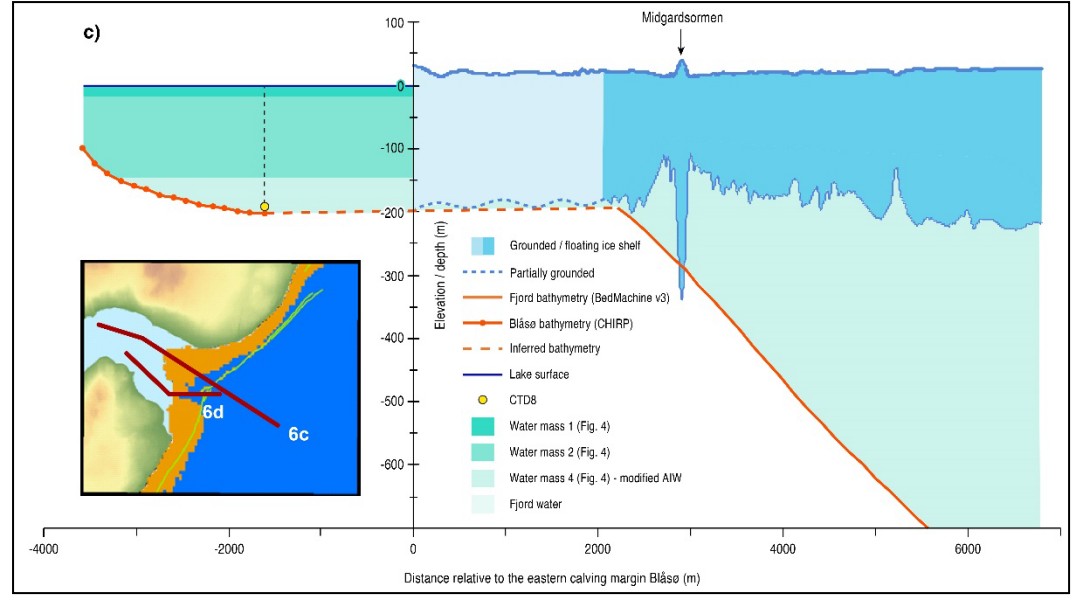

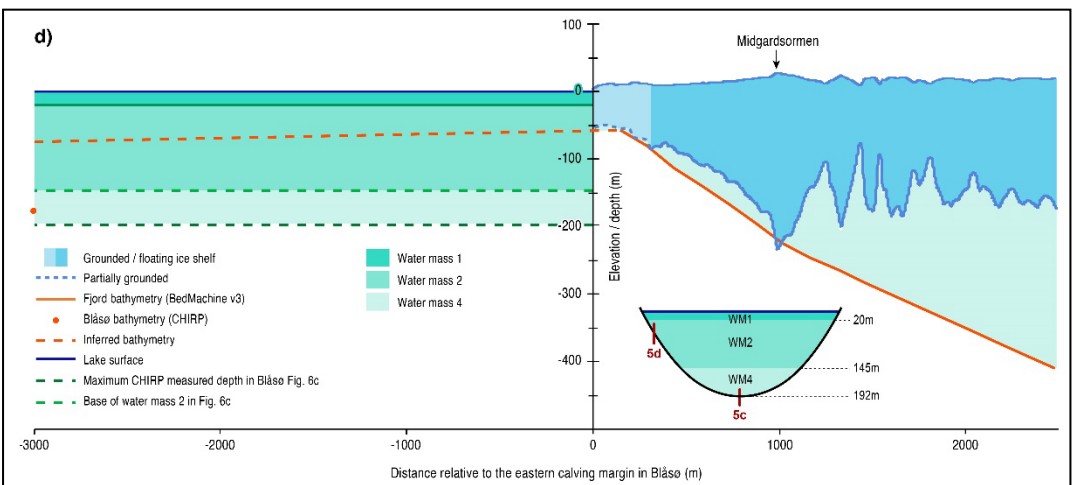

**Figure 6.** Hydrostatic analysis of the ice shelf calving margins in Blåsø. (**a**) Map showing the ice sheet and ice shelf surface as a grayscale hill shade from Arctic DEM. Inset boxes show locations of hydrostatic calculations of grounding shown for the western (**b**) and eastern (**c**) calving fronts in Blåsø., Panels (b), (c) and (d) show grounding line relationships in Blåsø, derived from hydrostatic analysis across the locations shown in inset panels. Profiles show the ice shelf surface (from the Arctic DEM), the ice shelf base (calculated from the ice surface elevation assuming hydrostatic equilibrium), and an inferred ice shelf base where the ice is no longer in hydrostatic equilibrium. The bathymetry of the fjord (BedMachine) and the lake (our data, fig 1) are also shown with an inferred bed between the lake and areas where hydrostatic analysis shows floating ice. (**b**) profile across the western margin of Blåsø where the maximum depth for the western basin (136 m) is constrained by CTD7. Despite being grounded for some distance between the lake calving margin and the fjord a temporary or periodic connection must exist to account for the brackish water present in water mass 3 and the dying fish observed in the western basin. Inset map shows hydrostatic analysis where orange indicates grounded ice and blue indicates floating ice. The western calving margin appears to be more grounded than floating (**c**) shows the eastern margin of Blåsø where the grounding line is defined by the Midgardsormen in the Arctic DEM. The pycnocline in the eastern basin of Blåsø and presence of AIW as water mass 4 implies an open connection between the lake

and the sub-shelf cavity must exist. The inset map shows hydrostatic analysis and that the eastern calving

margin is close to having a fully floating connection, associated with migration of the Midgardsormen (yellow

line) towards the northern side of the fjord. (**d).** Profile across the eastern calving front at the point where the

extent of grounded ice is narrowest (see inset panel for line of profile). The profile runs along the southern fjord

wall of Blåsø and so the ice shelf at the calving front is grounded in only ~50m of water. The profile shows the

depths of the deepest part of the eastern basin and the 145 m halocline taken from Fig 4 and projected onto this

section line.

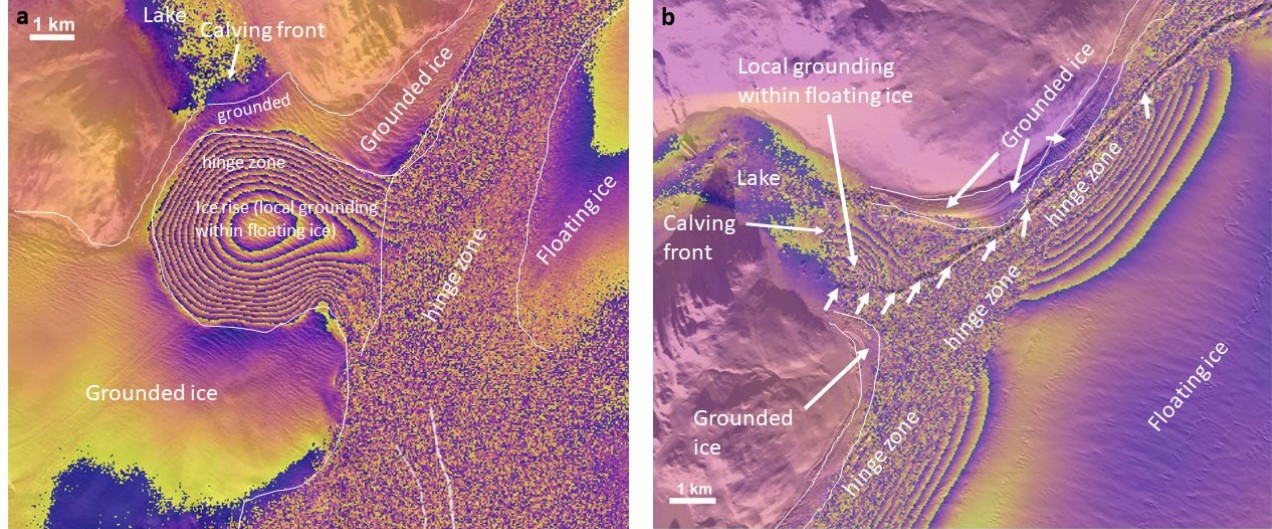

**Figure 7**. Interferometric analysis of calving margins in Blåsø. (**a**) Interferogram for western calving margin, and
(**b**) Interferogram for eastern calving margin. These show double differential interferograms from Sentinel-1 data
collected on 3, 9, and 15 April 2017, superimposed on a Sentinel-2 scene. Annotations show interpreted zones of
grounded and floating ice and the hinge zone. In b, short arrows show possible grounding of the Midgardsormen.

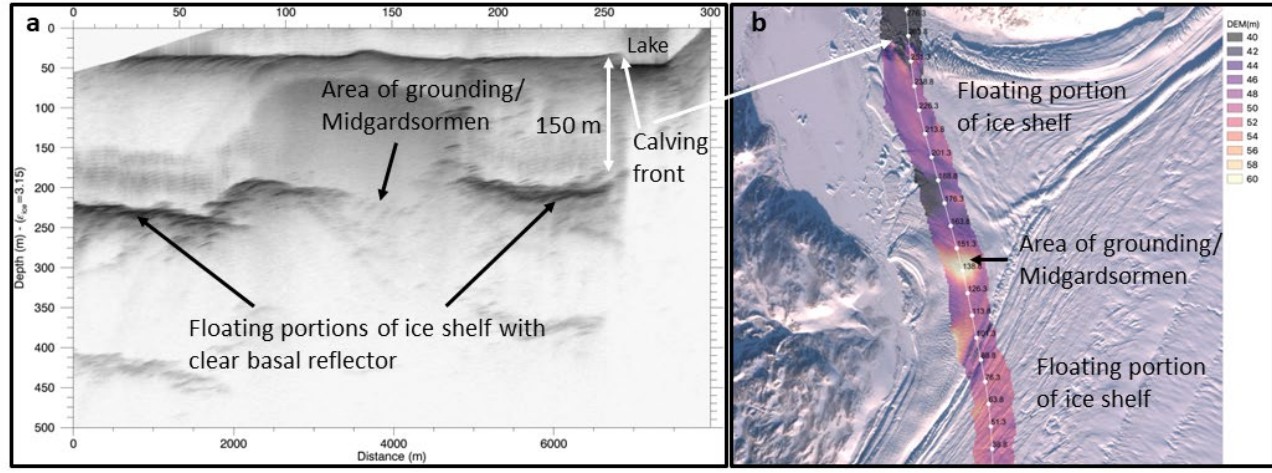

**Figure 8.** Airborne ice penetrating radar over the eastern calving margin in Blåsø. (**a**) Radargram from the AWI UWB multifrequency radar along the flightline shown in panel b. Clear basal reflectors and a zone of grounding (with elevated surface 'bump') are visible. The draught of the ice shelf, assuming a uniform velocity of radio waves in ice is ~150 m, closely coincident with the halocline depth measured at sites CTD5 and CTD 8. (**b**) Flight line of radar data, with colour shading of the elevation measured by the laserscanner. The area of grounding identified on interferometry is apparent in the DEM. Data acquired 14th April 2018.

700

**Table 1**. CTD locations in Blåsø. All times are UT.

| CTD profile ID | Latitude | Longitude | Max depth (m) | Date and time of CTD |
|---|---|---|---|---|
| CTD1 | 79° 35.738' N | 022° 35.596' W | 15 | 27/07/2017 15:59 |
| CTD2 | 79° 34.989' N | 022° 33.173' W | 10 | 31/07/2017 17:42 |
| CTD3 | 79° 32.017' N | 022° 30.523' W | 92 | 01/08/2017 17:25 |
| CTD4 | 79° 32.067' N | 022° 32.011' W | 126 | 02/08/2017 16:25 |
| CTD5 | 79° 36.303' N | 022° 05.251' W | 156 | 07/08/2017 21:35 |
| CTD6 | 79° 31.801' N | 022° 31.019' W | 134 | 08/08/2017 21:42 |
| CTD7 | 79° 31.774' N | 022° 30.523' W | 136 | 08/08/2017 22:08 |
| CTD8 | 79° 35.948' N | 022° 02.767' W | 192 | 10/08/2017 17:26 |

705