# Peer review of "Direct measurement of warm Atlantic Intermediate Water close to the grounding line of Nioghalvfjerdsfjorden (79°N) Glacier, Northeast Greenland."

_The Cryosphere, 2022_

## Referee Comment (RC1)

**Review:**

**2022GL098009: Direct measurements of warm Atlantic Intermediate Water close to the grounding line of Nioghalvfjerdsfjorden (79N) Glacier, North-east Greenland.**

**Overall Statement:**

Overall, this is an nice piece of work that uses diverse datasets to present important findings that relate to stability of the 79N Glacier and the Northeast Greenland Ice Stream. The manuscript is generally well written, the results are presented concisely, and the discussion and conclusions expand what is presented to establish their significance in the larger context. Below I have listed a set of comments that I believe will increase the accuracy and precision of the narrative, and will improve the clarity of the text. Most of these changes regard to presentation of the results and are not major. Ultimately, I think that this manuscript will make for a fine contribution to The Cryosphere and will advance the understanding of ice-ocean interactions around Greenland and the ice sheet's future evolution.

**Specific Revisions:**

I believe that it is typical to present "North-east" as "Northeast." I suggest changing this. throughout the text. Additionally, I see both northeast and NE throughout this text. I suggest to pick one expression and be consistent throughout.

I see that 79N, 79 N glacier, 79N glacier, and 79N Glacier are all used to refer to the same thing. Choose one shorthand expression and be consistent in its use throughout the text.

As Atlantic Intermediate Water is a primary focus of this manuscript, I suggest adding several sentences that discuss its larger origin, flow path (through Fram Strait), and depth range and temperatures on the NE Greenland continental shelf.

Both ice shelf and ice tongue are used to describe the floating portion of 79N Glacier. I know that there is some debate on what to call these features in Greenland based on their lateral constraints and geometries, but I think that it would help the manuscript to use one expression and be consistent in its use throughout the text.

Absolute Salinity units are presented as gkg$^{-1}$, g kg$^{-1}$, and g/kg. Choose the correct expression, which is g kg$^{-1}$ (with a space after the numbers, e.g., 34 g kg$^{-1}$), and correct this throughout the text.

Temperature units are presented with a space between the number and the unit. This is incorrect. Change this throughout the text to represent the correct notation which is, e.g., 4°C.

Distance units are presented with no space between the number and the unit, as well as with a space between the number and the unit. Choose one approach and be consistent throughout. I suggest to place a space between the number and the unit.

I do not think that presenting the different layers of the lake water column as discrete water masses is appropriate. Water masses refer to identifiable, discrete origins for the temperature and salinity range being observed. For instance, we know that Atlantic Intermediate Water derives from the North Atlantic and has a certain temperature and salinity range along the NE Greenland continental shelf and Glacial Meltwater is freshwater derived from melting glaciers. These are water masses. Please update the text to present the water column as having 3 or 4 layers, which are quasi well-mixed with a certain T, S range.

Figure 4 is referenced in the text before Figures 5 - 7. Generally, the figure numbering should reflect the order with which they are referenced in the text. Please either reference Figure 4 in the manuscript before 5 – 7, or renumber the figures.

I think that an opportunity has been missed to discuss the local input of fresh glacial meltwater into the lake from submarine melting of the ice faces that calve into the lake, as well as icebergs. This is not central to the main message of the paper, but the data were collected pretty close to the calving fronts so it would be nice to see a brief discussion of this mechanism added to the manuscript.

**Background and rationale:**

Li 33: I suggest to be more precise with this statement and change it to:

"from the NE sector of the ice sheet to Fram Strait."

Li 34 – 35: Suggest to change to "NEGIS and the ice shelves that extend from its margin"

Li 34 – 36: Provide a reference to support this statement.

Li 38. – 39: Does this citation state that ocean temperatures (or thermal driving) will double by 2100 or the rate of ocean warming will double by 2100? Update this sentence to clarify this.

Li 44 – 40: Rewrite these sentences to improve their structure and more clearly introduce the study region to the reader. Please consider the following suggestions during rewriting:

- State simply that NEGIS extends from the ice divide to the coast
- State the flow speed range from the onset (slow) to the coast (max rate)
- State approximately where the three outlet glaciers split off from one another and then name them. Reference Figure 1 at this point.
- Introduce more clearly the bed geometry near the coast.
- Introduce the 79 N Glacier ice shelf, mention its flow direction, length, GL depth, fjord depth, and thickness range (which is ~100 – 600 m – fix this and add some references).

Li 47 – 48: Suggest to change "is front by an ice shelf" to "extends in an ice shelf"

Li 59: Suggest to change to "79 N Glacier and its ice shelf" or "79 N Glacier ice shelf."

Li 60 – 61: Add melt rate estimate from Wilson et al. (2017). See the full reference below under the References section.

Li 63: Reference Figure 1 after the calving front statement.

Li 70: Was AIW found within the ice shelf rift or beneath it? I suppose there was almost certainly some mixture of AIW in the water column within the rift, but for the context of this introduction, where AIW = heat, it might be better to say beneath the ice shelf rift.

Li 72: Again, I believe all the instruments were beneath the ice shelf base, so the statement "A record from an Ice Tethered Mooring (ITM) situated in this rift" seems misleading to me.

Li 73: It could be worthwhile to mention the substantial heat throughout the water column at the ITM site, primarily due to only weakly diluted AIW.

Li 74 – 79: I suggest adding "meltwater-enriched" to "outflow" at some point in these sentences to more clearly communicate the sub-ice overturning circulation to non oceanographers.

Li 80 – 81: These sentences would benefit from the aforementioned suggested introduction to AIW.

Li 83: Suggest to change to "increasingly, warmer, more saline, and shoaling AIW layer"

Li 85 – 86: I believe that is what this paper was saying, but be sure to mention somewhere in here that this is an increase in the "overall" or "average" ice shelf melt rate.

Li 89 – 90: I do not recall if Mayer et al. (2018) set out to estimate a thinning rate over the whole ice shelf. If so, state that explicitly with the thinning rate range. If not, state explicitly that this thinning rate of up to 12 m a$^{-1}$ is local only to Midgardsormen. The reader will be confused otherwise, as the Rignot and Jacobs (2002) approach suggests an increase in melting of 5 m a$^{-1}$, which can be equated to thinning if we ignore large changes in ice dynamics. This is less than half of 12 m a$^{-1}$, which is a significant difference.

Li 98: Suggest to change "then this should have profound" to a more precise statement such as, "then changes in its thermohaline properties should have profound."

Li 100: Suggest to change "flux, extent, properties and interaction of AIW with the floating ice and grounding line" to "delivery of AIW to the sub-ice shelf cavity and the degree of thermodynamic interactions with the ice shelf base" to improve sentence readability.

Li 102 – 104: Poorly written run-on sentence. Rewrite to improve readability.

Li 106: Add the distance of Blasø from the grounding line, and reference Figure 1.

Li 107: Correct to Interferometric synthetic Aperture Radar (InSAR).

**Study Area:**

Li 111: Is this sentence correct? Milne epishelf lake in Canada is one that comes up often in the literature that is between an ice shelf and a lodged ice mass in a fjord with fjord walls on its sides. That is, it is not strictly bounded by an ice free land area and an ice mass. Please take a look through the literature to make sure that this statement is correct.

Li 111 – 112: Aren't all epishelf lakes freshwater on above the ice draft and seawater below? Isn't freshwater typically considered necessary for a body of water to be considered a lake? I suggest to correct these two sentences then combine them into one cohesive sentence that accurately defines an epishelf lake.

Li 114: Why must the ice be in hydrostatic equilibrium for its underside to determine the transition to seawater? Wouldn't the underside depth determine the onset of seawater regardless of the degree of flotation?

Li 115 – 119: Can you expand this thought a little further to explain concisely how epishelf lakes have been used to infer past glaciological change?

Li 122 – 123: Would the southern lake margins also receive freshwater from submarine melt of the ice faces and summertime runoff?

**Methods:**

Please add subsection headings to improve the organization of this section.

Li 138 – 140: The structure of this sentence makes it unnecessarily difficult to read. Suggest to rewrite the sentence with active voice to more clearly and concisely communicate the idea.

Li 145: Please provide a reason for the depth differences. Perhaps they are within the instrument uncertainties at these depth ranges? If so, state this uncertainty range explicitly.

Li 148 – 149: Please convert the C, T, and P uncertainties to Conservative Temperature and Absolute Salinity uncertainties, as these are the properties that the hydrographic data should be presented in.

Li 146 – 156: Were the CTD post-processed at all? If not, I suggest to post-process the data to improve their quality, as this is standard procedure in oceanography. This webpage explains nicely how to post-process the profiles: https://docs.rbr-global.com/rsktools/files/latest/57311819/57311821/1/1593023510371/PostProcessing.pdf.

Li 159: Please convert this uncertainty to a vertical range based on the observations.

Li 162: Cite Figure 1 b at the end of this sentence.

Li 174: Is this the annual DEM for 79 N for 2017 or is it a single DEM spanning multiple years? Please add this information to the text.

Li 181: See above comment on correcting InSAR acronym definition.

Li 184 – 185: I see the reference to the full InSAR processing method, but it would be nice to know in the text if tidal elevation data necessary for the vertical correction applied? If they are, did you extrapolate your measurements back in time with a harmonic analysis or use the Dansmarkhan tide gauge? Did you use the CATS2008 tide model?

**Results:**

Please add subsection headings to improve the organization of this section.

Li 205 – 206: Be consistent with units. Either choose cm or m for tidal range. Figure 2 presents amplitude as cm, so perhaps that is the best route forward.

Li 208 – 211: I do not understand the point of this text. Is it just saying that the CTD profiles were retrieved within some range of the seafloor? I think that most of this is dead text that can be cut out so that this paragraph can be combined with the next to more concisely state the results.

Li 212: Replace "haloclines and accompanying thermoclines" with "pycnoclines where temperature and salinity increased rapidly with depth."

Li 225 – 226: Is this sentence referring to the upper limit of the pycnocline or its thickness? This is not clearly communicated with this sentence.

Li 227 – 229: This is a very interesting finding that in its current form is kind of a distraction from the narrative. Please add a temperature and salinity range where these fish have been observed to live in to fix this. Also, this is minor, but were the fish dying or dead? If they were interpreted as dying, what was the reasoning for this interpretation. Please write more precisely.

Li 233 – 235: Poorly written run-on sentence that contains multiple thoughts. Please rewrite as two sentences.

Li 243: Correct to "free floating (Fig. 5d)."

Li 263: Suggest to change to "ice penetrating radar data," because there are many different frequencies of radar used to measure ice and some do not penetrate through the whole ice column.

Li 285: Please convert these data ranges to months so that the reader can more easily compare the lags presented in this manuscript to those in Wilson and Straneo (2015) and Schaffer et al. (2020).

**Discussion:**

Li 273: Please change "melt-mixing line" to "meltwater mixing line (Gade, 1979)." I've added the citation to the reference section below.

Li 273: Since this is the first mention of glacial modification of AIW through melt input, I suggest to clarify this as "glacial melt modified AIW" or "glacially modified AIW" and to clarify that this is colder and fresher (less thermal driving) than pure AIW.

Li 274: Again, the ITM was deployed through a rift, but all the sensors resided beneath the ice shelf base, so I think "deployed in the rift" is slightly misleading.

Li 276: The expression "50 km inboard of the calving front" is somewhat awkward. Consider using a different word such as "upstream, upglacier, westward" or something similar.
Li 283: The reference to the Schaffer et al. (2020) paper is awkwardly placed. I suggest to place it after "grounding line."

Li 296 – 299: Poorly written 59 word run-on sentence containing at least two ideas. Please rewrite this sentence and break it into two shorter sentences.

Li 300: I suggest to use "pycnocline" instead of "halocline", because both temperature and salinity characteristics are referred to here.

Li 302 – 303: Why does the 145 m pycnocline have to be a proxy for the ice shelf draft "in hydrostatic equilibrium?" Why can't it simply reflect the ice shelf base depth?

Li 300 – 305: There is another fresh water mass that is not considered here that will fill the lake. That is glacial meltwater from submarine melting. Please update this paragraph to reflect this.

Li 306 – 309: I do not think that this is the proper mechanism to explain the 5 m difference in pycnocline depth considering the insane vertical stratification across this feature. Internal wave amplitudes decrease with stratification and their speeds increase, so I would expect them to be much smaller and quite fast. I would expect that it is more likely that the seawater layer simply goes up and down with the tide, then there is quite a strong pressure driven flow into and out of the lake where water parcels flow down isopycnals. Please expand the discussion to reflect this comment or include a rebuttal, with a scaling argument in the text, that defends the internal wave interpretation.

Li 322: Is the tidal exchange unencumbered? Earlier in the text it was hypothesized that the somewhat reduced tidal amplitudes results from the partially grounded margin of 79N Glacier inhibiting tides. Please clarify this.

Li 337: Please change to "dying marine fish" so that the reader knows at this point, not later, that Arctic Cod are strictly marine. Also, see prior comment above that points out that it would be helpful to state the thermohaline range that these fish live in.

Li 3338: The statement that there is "no seawater present in the deepest parts of the western basin" is incorrect, because brackish water contains seawater. Please correct this portion of the text to be more precise.

Li 344: Ok, so this is way outside of my area of study, but wouldn't fish that live in denser seawater and therefore balance their buoyancy to the denser water, sink if placed into less dense brackish water?

Li 350 – 352: Ok, I do not disagree that 79N Glacier appears to be thinning significantly, but how do several CTD profiles from a single year that reveal seawater in the lake show this? They cannot resolve change because they are from a single point in time – they just show that there is seawater at depth in the lake. The migrating margin I believe shows this, but this is not explained sufficiently here, and I don't think these data are really shown in this manuscript. Please expand this discussion to more sufficiently defend the reasoning that Lake Blasø can tell us about the thinning 79N Glacier. I believe that this will require at least an additional paragraph, perhaps two.

**Conclusions:**

Li 359 – 360: It is important to clarify that the AIW at the ITM site that is being referenced has already been glacially modified. At least this is what I thought was communicated earlier in the text. Either way, it would be helpful to clearly communicate the temperature and salinity difference between the AIW observed in Lake Blasø and the warmest and most saline AIW observed at the ice shelf front. The Schaffer et al. (2020) or Lindeman et al. (2020) reference should provide the necessary information.

Li 373 – 377: This statement ignores the logistical difficulties of establishing a continuously-monitoring mooring in the western basin of Lake Blasø, which will be significant. Calving icebergs from the tidewater fronts will likely have a draft close to the maximum depth of the fjord and will have a high likelihood of running into the mooring. If this statement is to be left in the text, then there should be a disclaimer about the inherent risk in establishing a mooring in the western basin.

Li 881: Suggest to change to "glacial meltwater from basal melting of the ice shelf."

**Figures:**

Figure 1: Consider adding a reference map that places NEGIS in context of Greenland as a whole, and adding glacier demarcations to the inset figure. I believe these data can be downloaded here http://imbie.org/imbie-3/drainage-basins/. Add a northward-pointing arrow or longitude and latitude lines to panel A. It would be nice to have several more ticks on the color bar to more easily identify lake depths in relation to the color scale.

Figure 2: Perhaps it would be clearer to present these data as deviations about a 0 cm elevation. That would make the data easier to compare and would represent tidal fluctuations more accurately.

Figure 3: Please label x axes as Conservative Temperature ($\Theta$) and Absolute Salinity ($S_A$)

Figure 4: Please label axes as Conservative Temperature (Θ) and Absolute Salinity (S$_A$). Also, The caption says that the blue data are from the Eastern Basin, but the legend labels them as the Western basin. Please correct this.

Figure 5: This is a very informative figure, but in its present form it takes up 2.5 pages and requires some improvement. I suggest the following changes:
- I suggest to either split into three figures or remake what is currently Figure 5 so that it all fits on one page.
- Additionally, the vertical scales on d – f) are different. It would improve the interpretation of this figure if these panels all had the same vertical scale or the panels heights varied with respect to the vertical scale.
- The inclusion of a filled space beneath the ice shelf that is either modified AIW or fjord water is misleading. The authors do not data to prove this. Please either label this part of the figure as AIW (interpreted) or something similar. In reality the sub-ice shelf water column will have up to 4 different water masses mixed into it, with AIW probably being the dominant water mass.
- I suggest to interpolate between multiple CTD profiles to fill in the Lake Blaso water column structure.
- Finally, I suggest to replace the multiple legends with a single legend and nest the maps of the lake locations in the empty space to improve the figure.

**References:**

Wilson, N., Straneo, F., & Heimbach, P. (2017). Satellite-derived submarine melt rates and mass balance (2011–2015) for Greenland's largest remaining ice tongues. *The Cryosphere*, *11*(6), 2773-2782.

Gade, H. G. (1979). Melting of ice in sea water: A primitive model with application to the Antarctic ice shelf and icebergs. *Journal of Physical Oceanography*, *9*(1), 189-198.

---

## Referee Comment (RC2)

I apologize to the authors and journal editors for being late with this review. I got carried away with other responsibilities but thoroughly enjoyed reading the manuscript and appreciate the opportunity to comment on it.

This manuscript blends geophysical, remote sensing, hydrographic, glaciological, oceanographic and limnological datasets together to 1) demonstrate conclusively that Blåsø, a fresh/brackish body of water at the ice shelf margin, is an epishelf lake and to 2) argue that Atlantic Intermediate Water (AIW) can reach the grounding line 79N glacier and is interacting with the ice there. The paper is well presented and of significance to the readership of this journal. I have no hesitation recommending it be accepted for publication in The Cryosphere as long as some minor comments/suggestions are addressed.

Derek Mueller, Carleton University [2022-12-17]

**General comments**

*Epishelf lake:*
To my knowledge this paper is the first to describe Blåsø as an epishelf lake. These lakes are rare and unique so it was somewhat surprising that the significance of this was not highlighted as much as it could have been.

Gibson and Anderson (2002) was cited and it might be a good idea to explain examine Blåsø within the framework they illustrated in their Figure 2 where there are two types of epishelf lakes – Type 1, "with freshwater directly overlying marine water" (I assume this is the case for the east basin) and Type II – "with indirect connection to the marine environment" (perhaps a more suitable description for the west basin – if there is a conduit on that side?). There could also have been more description of Blåsø. How big is the catchment? How much of it is glacierized? How common is summer ice cover? Is there more to say about water mass 2 in the eastern basin and water mass 3 in the western basin? How did they form and how does they persist?

You write [for a Type 1 epishelf lake] "the depth of the transition between marine and brackish/fresh water is controlled by the draught of the floating ice" but this should really be the *minimum* draft [draught] of the ice shelf (whether this point is in local hydrostatic equilibrium or not). A caveat here is that epishelf lakes can over-deepen in the summer due to freshwater input (see Hamilton et al., 2017 and Bonneau et al. 2021).

As as consequence of the above, it is challenging to find the minimum draft of the ice shelf that is controlling the outflow. The radar transect presented in Fig 7 is likely the best approach available, far better than estimating draft using hydrostatic equilibrium. So, I agree that the airborne radar (and InSAR) are more reliable (stated on ms line 328), although it is fine to include all the data for context. The fact that the minimum draft you highlight (150 m) is so close to the interface between water mass 2 and 4 is pretty convincing (but also see below).

*Uncertainty/Errors:*
It would be helpful to know more about the uncertainty of the various datasets that are used in the analysis as most of the data is presented without any associated error bars. The propagation of errors and comparisons of the datasets can then be discussed. For example:

- The water mass 2 - 4 interface varied by 5 m (n=2) so how well is this constrained?

- How are bedmachine data produced?  What error is associated with this product under grounded and floating ice?
- Is it reasonable to follow Morlighem et al., (2017) and assume an ice density of 917 kgm-3 and sub-shelf water density of 1023 kgm -3?  You may not have any ice density measurements on hand (not many do), but is this water density realistic, given data from Figure 4, water mass 4?
- What about DEM precision and accuracy?  How does this and density uncertainty impact the HE calculations?
- What is the error in the radar thickness/draft and InSAR?

*Oceanography and ice-ocean interactions:*
I have not read Lindeman et al. (2020) but I feel there are probably answers to the following questions within it.  These could be brought into the text so that it is clear in your manuscript without readers having to go elsewhere to find info.

There is mention of AIW being present at 500 m in the rift mooring data but I have the sense from the description that this varies over time.  Please explain the dynamic nature of the AIW – what depths is it found at, what range in salinity and temperature, what is above (and below, if relevant) this water mass?.  It would be helpful if this explanation also cleared up why you match Blåsø water properties to the ITM data at a specific moment in time.  How consistent is this water at the rift ITM?

Figure 4 shows mixing lines from the 500 m level at the ITM mooring at 3 times of year and some daily values (unclear what time of year they are from).  These melt lines align with water mass 4 properties and this is used to infer that AIW interacted with the ice shelf at some depth (presumably between 500 m and ~200 m).  But are there any other water masses or combinations of water masses that could account for the water properties seen year?  In other words, are there any alternative hypotheses to explore before espousing your interpretation that AIW is the culprit?

To tie together the above 2 paragraphs, if you had a profile in the east basin from January 2018 would you hypothesize that it would be on the July 2017 melt line?

Lastly, I think adding bathymetric contours to the map in Figure 1 or elsewhere and an along fjord cross-section of the bathymetry, and ice draft and elevation to complement text on lines 45-50 and 60-70 would be very helpful.

**Detailed minor comments**

line 35 - exhibited little response to atmospheric and oceanic warming [in or over] the decades

line 38 - Model projections suggest that ocean warming around Greenland will double.  Do you mean the rate will double or the temperature relative to some reference period?  Explain

line 47 - grounding line (~600 m below sea-level).  This doesn't seem to match the ice shelf thickness of 300 to 100 m.

line 88 – Midgardsormen ridge description is unclear here.  Why did it flow backward?  How much landward migration was there.  This feature becomes clearer later in the text but it would help to have a better explanation here.

line 104-108 – You could include characterizing Blåsø as a specific purpose of your paper in the statements here

line 107 - synthetic Aperture Radar Interferometry  (capital S)

line 113 - The depth of the transition between marine and brackish/fresh water is controlled by the [minimum] draught of the floating ice.  This is true if the adjacent ice shelf is not grounded (whether or not it is perfectly in HE or not).

line 143 - CHIRP (Compressed High Intensity Radar Pulse)  - I am really unclear on this.  Does radar actually work underwater?  Are you not using sonar?

line 148 – For pressure is that +/- 0.05% of the pressure value or the full scale – if the latter, please share what that is.

line 159 – please give full scale or convert to accuracy in pressure units

line 225 34.4 to 34.7 g/kg.  change to gkg-1 to be consistent

line 276 – 79N Glacier [capital G]

line 315 – thank you for explaining the 5 m discrepancy – It is true there are internal waves in Milne Fiord epishelf lake. I certainly don't recall them being on the order of 5 m.  Maybe there is another explanation for this?

line 317 79N Ice Shelf – proper name capitalization.

line 357 – replace measurement x1 with a synonym to avoid redundant text

line 363 – can you give error/uncertainty here?

Figure 1A
Would it be possible to see a bit further to the west?
Would it be possible to add some bathymetric contours to this figure?
Why are there 2 ESA CCI 2017 grounding lines?

Figure 1B
Would it be possible to krig and contour the bathymetry in addition to the CHIRP data?  The colour ramp could use some intermediate values between 0 and 212 m.

Fig 1 caption
Red dots show where moorings have measured [water] flow direction [no s – only one arrow] in (yellow arrow)
SG = Storstrømmen Glacier – remove comma

Figure 2 – the grey line is hard to see.
How far away is the Dansmarkhan tide gauge?

Each of the 3 records could be centered on the tide gauge data by offsetting by the difference in the average values of coincident records.  They would still have arbitrary datums but would be aligned.

Figure 4a  caption
The shaded range indicates the instrument accuracy  - do you mean precision?
Daily values -from ITM – are these for specific times of year or for the entire record?

Figure 5a
Unclear what the red dashed line is
The dashed grounding line are 2 mutually exclusive options for how the GL could go?  It should be clear.
Midgarsormen line is green in the figure and yellow in the legend.
Need a legend for the toppography (colours)
Note the inset maps c and d might be better placed in the cross sections under 5d, e and f.  (along with the transect).  There should be space for them there under the lake – with their own scale bar too.  The legends can move over to the right

Fig 5e – there is no partially grounded dashed purple line
Fig 5f – explain the 6c an 6b lines

Figure 5 caption -
Midgardsormen (yellow line)  is green
eastern calving front at the point where the extent of grounded ice is narrowest (see Fig 5).  - but this _is_ Figure  5 – do you mean somewhere specifically?

Figure 6
In E, short arrows show possible grounding of the Midgardsormen.  There is no E

Figure 7
The numbers in b are very small and hard to read

Table 1 – need to add degree symbol and minute symbol

**Citations I made that are not already in the manuscript**
Bonneau, J., Laval, B. E., Mueller, D., Hamilton, A. K., Friedrichs, A. M., and Forrest, A. L.: Winter dynamics in an epishelf lake: Quantitative mixing estimates and ice shelf basal channel considerations, J. Geophys. Res. Oceans, 126, e2021JC017324, https://doi.org/10.1029/2021JC017324, 2021.

---

## Community Comment (CC1)

**Comments on *Direct measurement of warm Atlantic Intermediate water close to the grounding line of Nioghalvfjerdsfjorden (79N) Glacier, North-east Greenland**

**Global statement**

Interesting and timely results of Atlantic water close to the grounding line of 79N. I think this manuscript should be accepted after the comments of the two reviewers are addressed. I also have some minor comments that I believe would improve the manuscript.

**General comments**

Neither An et al .(2021), Lindeman et al. (2020), Mayer et al. (2018) or von Albedyl et al. (2021) present a full map of the ice shelf (glacier tongue) thickness. I feel since you are already using ArcticDEM for areas near the epishelf lake, you could present an ice thickness contour map of the ice shelf. That would not be too much work if you already have the grounded ArcticDEM data. Maybe that would help explain the weird circulation pattern outflowing through Dijmphna Sund.

I did not see any discussion about subglacial discharge. I think it is warranted to at least briefly discuss why it is not relevant in this study. You are really close to the grounding line and your CTDs are taken in summer, this is prime location and timing to see subglacial discharge. You mention that a "simple plume model would account for the thinning […]". Subglacial discharge is sometimes included in these plume models, maybe that's a place to emphasize subglacial discharge does not matter so much in this case.

I agree with reviewer 1 comment that using "water mass" is misleading.

I would add a few sentences on why "water mass 2" in the eastern basin is fresher than "water mass 3" in the Western basin. Isn't the eastern basin more connected to the ocean than the western basin?

**Edits**

L36: "Unlike many other sectors of the Greenland Ice Sheet, NEGIS and the ice shelves that front it exhibited little response to atmospheric and oceanic warming for the decades immediately prior to the mid 2000s." Maybe add the citation(s) to this statement directly at the end here instead of after the next sentence.

L46: "NEGIS flows at ~1200 m a-1 and upstream from the 79N Glacier grounding line (~600 m below sea-level) the basin floor deepens to ~1000 mbsl (Bamber et al., 2013)". Is that velocity at the grounding line of 79N? Maybe divide this sentence in two, one about the velocity (with where the 1200 m a-1 velocity is taken) and one about the bathymetry.

L56: "Humbert et al (submitted) also identify a recent shift in calving style and fracturing at the calving front of 79N Glacier." A shift from which style to which style? Maybe also add a sentence on the implications of this statement.

L59: "Based on the rapid decrease in thickness of the ice shelf, to only 330 m within 5 km of the 79N Glacier grounding line ,…"  To 330 m, ok, but from what thickness?

L69: "There is only one measurement of AIW in the cavity, where it has been detected in a rift in the 79N ice tongue, located ~10 km behind the calving front" When was this measurement? Is this from the ITP? If so make this clearer.

L96: "groundling" [typo]

L98: "If AIW is circulating throughout the cavity beneath the floating portion of the 79N Glacier then this should have profound consequences for stability of the grounding line (An et al., 2021) and the ice shelf." Change "floating portion" by ice shelf or glacier tongue and be consistent throughout the paper, i.e. no alternating between both.

L112: "Where a source of freshwater feeds into the lake [add coma] a salinity-driven stratification forms with the more saline marine layer capped by a freshwater layer."

L138: "As part of a wider programme to characterise and sample water and sediments in Blåsø, to understand past changes in the 79N Glacier, the bathymetry of the lake was mapped and multiple CTD profiles were measured in different parts of the lake." Sluggish, maybe break into two sentences?

L146: While at it, what's the sampling rate of the CTD and how fast was it lowered? How was the data averaged/binned?

L175: Partially out of personal interest, but likely relevant to many others: did you compare bedmachine3 to your CHIRP survey? If so, one or two sentences on the comparison would definitely be relevant.

L208-211: Use either just CTD or just CTD profile or water profiles, but be consistent.

L227: "During bathymetric and CTD surveys [add coma] we observed …"

L308: "It is more likely that the difference is caused by internal waves which can be created where tidal currents drive water parcels, especially, on steep slopes (Munk and Warren, 1981)". You mean baroclinic tides? Any evidence of baroclinic tides in the ITM or other moorings record? This is a pretty big density jump, 5 m internal waves would be quite impressive. Could it be just that what you call water mass 2 is draining away? (making its way out through cracks and channel). Seasonal cycles of deepening and shoaling have been reported in Milne Fiord epishelf lake (Hamilton 2017, Bonneau 2021).

L316: Again, possible internal waves, but I would not conclude it is without a doubt based on two CTD profiles. High frequency internal waves in Milne Fiord epishelf lake have a maximum amplitude of 15 cm and a period of 50 min.

Figure 1A: Would it be possible to add some bathy contours from bedmachine3? Need a scale for the velocity arrows.

Figure 1B: I think you have enough data to generate a decent bathy map of the lake. It would be nice see that. And overlay the data points?

Figure 3, 4: You say you are using conservative temperature and absolute salinity. These should be your labels.

Figure 4: Are the dots your CTD data points and the lines just link the data points? If so I would remove the lines and make the markers a little larger, as it is usually done on T/S diagram.

Figure 5D: Check y axis, something is wrong.

Table 1: Not sure it brings something to the paper, could easily do without.

---

## Author Comment (AC1)

**Review:**

**2022GL098009: Direct measurements of warm Atlantic Intermediate Water close to the grounding line of Nioghalvfjerdsfjorden (79N) Glacier, North-east Greenland.**

**Overall Statement:**

Overall, this is an nice piece of work that uses diverse datasets to present important findings that relate to stability of the 79N Glacier and the Northeast Greenland Ice Stream. The manuscript is generally well written, the results are presented concisely, and the discussion and conclusions expand what is presented to establish their significance in the larger context. Below I have listed a set of comments that I believe will increase the accuracy and precision of the narrative, and will improve the clarity of the text. Most of these changes regard to presentation of the results and are not major. Ultimately, I think that this manuscript will make for a fine contribution to The Cryosphere and will advance the understanding of ice-ocean interactions around Greenland and the ice sheet's future evolution.

***Thanks for the review and the positive comments on what we have done to draw together diverse datasets to say something important about the stability of 79N Glacier.***

**Specific Revisions:**

I believe that it is typical to present "North-east" as "Northeast." I suggest changing this. throughout the text. Additionally, I see both northeast and NE throughout this text. I suggest to pick one expression and be consistent throughout.
***We will change this as suggested.***

I see that 79N, 79 N glacier, 79N glacier, and 79N Glacier are all used to refer to the same thing. Choose one shorthand expression and be consistent in its use throughout the text.
***We will change this as suggested.***

As Atlantic Intermediate Water is a primary focus of this manuscript, I suggest adding several sentences that discuss its larger origin, flow path (through Fram Strait), and depth range and temperatures on the NE Greenland continental shelf.
***We will add a short explanation and some key references to help introduce AIW***

Both ice shelf and ice tongue are used to describe the floating portion of 79N Glacier. I know that there is some debate on what to call these features in Greenland based on their lateral constraints and geometries, but I think that it would help the manuscript to use one expression and be consistent in its use throughout the text.
**Both exist freely in literature; we agree consistency may help and so have elected to refer to the floating portion as 'ice shelf' throughout**

Absolute Salinity units are presented as gkg$^{-1}$, g kg$^{-1}$, and g/kg. Choose the correct expression, which is g kg$^{-1}$ (with a space after the numbers, e.g., 34 g kg$^{-1}$), and correct this throughout the text.
***We will change this as suggested.***

Temperature units are presented with a space between the number and the unit. This is incorrect. Change this throughout the text to represent the correct notation which is, e.g., 4°C.
**Opinions on this vary, including the International Bureau of Weights and Measures who use a space, but many publishers who do not. We don't mind so will adjust to fit the preference of the reviewer.**

Distance units are presented with no space between the number and the unit, as well as with a space between the number and the unit. Choose one approach and be consistent throughout. I suggest to place a space between the number and the unit.
**Apologies we missed some numbers with no spaces to their units. We will change this as suggested.**

I do not think that presenting the different layers of the lake water column as discrete water masses is appropriate. Water masses refer to identifiable, discrete origins for the temperature and salinity range being observed. For instance, we know that Atlantic Intermediate Water derives from the North Atlantic and has a certain temperature and salinity range along the NE Greenland continental shelf and Glacial Meltwater is freshwater derived from melting glaciers. These are water masses. Please update the text to present the water column as having 3 or 4 layers, which are quasi well-mixed with a certain T, S range.

**The water column in the CTD casts in the eastern (western) basins is consistent with a 3(2) layer structure with large density gradients in between the layers. But these are not homogenous layers (i.e. not slab-like, as the term 'layer' would imply to us) and the properties of the same 'layer' can plot in different parts of Temperature-Salinity space depending on where in the lake the CTD cast is taken from. So we prefer the term water mass. To address the concern of the reviewer we will clarify that we are only referring to the water in the lake, and will include a summary in discussion that explains the likely origins of the water, along these lines:**

**The top layer is a fresh layer likely from runoff or above the surface melt of floating ice whose temperature is determined by solar heating. This layer is confined to the top ~20m which also happens to be the depth of the sill that confines the western basin, so it can spread between the different basins unimpeded. The properties are quite similar within all basins although the temperature and salinity of the topmost few metres varies according to freshwater input, lake ice melting and proximity to calving fronts.**

**The intermediate layer is a brackish watermass whose properties are distinct between the eastern and western basin (which, is consistent with the presence of at least the 21 m sill which keeps the east and west basin waters within this layer separate. Likely this watermass is affected by the iceshelf/icebergs that confine it. Its origins are unclear but wintertime vertical mixing of deeper (saltier) water masses could play a role.**

**The 4th layer is the modified AIW hence a watermass.**

Figure 4 is referenced in the text before Figures 5 - 7. Generally, the figure numbering should reflect the order with which they are referenced in the text. Please either reference Figure 4 in the manuscript before 5 – 7, or renumber the figures.
**Fig 4 is called at line 213, Fig 5 at line 232. They are called in order.**

I think that an opportunity has been missed to discuss the local input of fresh glacial meltwater

into the lake from submarine melting of the ice faces that calve into the lake, as well as icebergs. This is not central to the main message of the paper, but the data were collected pretty close to the calving fronts so it would be nice to see a brief discussion of this mechanism added to the manuscript.

*We will include a short description of the different sources/inputs of freshwater to the lake including submarine melting of the ice faces, fluvial input from the delta and other streams, supraglacial melt flowing off the ice shelf, and melting of lake ice.*

**Background and rationale:**

Li 33: I suggest to be more precise with this statement and change it to:
"from the NE sector of the ice sheet to Fram Strait."
*This section (lines 33-59) will be slightly reworded to address multiple comments of Reviewers 1 and 2*

Li 34 – 35: Suggest to change to "NEGIS and the ice shelves that extend from its margin"
*This section (lines 33-59) will be slightly reworded to address multiple comments of Reviewers 1 and 2*

Li 34 – 36: Provide a reference to support this statement.
Li 38. – 39: Does this citation state that ocean temperatures (or thermal driving) will double by 2100 or the rate of ocean warming will double by 2100? Update this sentence to clarify this.
*This section (lines 33-59) will be slightly reworded to address multiple comments of Reviewers 1 and 2*

Li 44 – 40: Rewrite these sentences to improve their structure and more clearly introduce the study region to the reader. Please consider the following suggestions during rewriting:

- State simply that NEGIS extends from the ice divide to the coast
- State the flow speed range from the onset (slow) to the coast (max rate)
- State approximately where the three outlet glaciers split off from one another and then name them. Reference Figure 1 at this point.
- Introduce more clearly the bed geometry near the coast.
- Introduce the 79 N Glacier ice shelf, mention its flow direction, length, GL depth, fjord depth, and thickness range (which is ~100 – 600 m – fix this and add some references).

*This section (lines 33-59) will be slightly reworded to address multiple comments of Reviewers 1 and 2*

Li 47 – 48: Suggest to change "is front by an ice shelf" to "extends in an ice shelf"
Li 59: Suggest to change to "79 N Glacier and its ice shelf" or "79 N Glacier ice shelf."
*This section (lines 33-59) will be slightly reworded to address multiple comments of Reviewers 1 and 2*

Li 60 – 61: Add melt rate estimate from Wilson et al. (2017). See the full reference below under the References section.
*OK*

Li 63: Reference Figure 1 after the calving front statement.
*OK*

Li 70: Was AIW found within the ice shelf rift or beneath it? I suppose there was almost certainly some mixture of AIW in the water column within the rift, but for the context of this introduction,

where AIW = heat, it might be better to say beneath the ice shelf rift.

*__We will reword the description of the Ice-Tethered Mooring (ITM) to clarify that the sintrument was deployed from (and tethered in) the rift but the mooring string sampled the water below the ice shelf, not in the rift itself.__*

Li 72: Again, I believe all the instruments were beneath the ice shelf base, so the statement "A record from an Ice Tethered Mooring (ITM) situated in this rift" seems misleading to me.

*__See above__*

Li 73: It could be worthwhile to mention the substantial heat throughout the water column at the ITM site, primarily due to only weakly diluted AIW.

*__This feels implicit from the existing wording and is a potential distraction from the focus of the paper in Blaso__*

Li 74 – 79: I suggest adding "meltwater-enriched" to "outflow" at some point in these sentences to more clearly communicate the sub-ice overturning circulation to non oceanographers.

*__OK__*

Li 80 – 81: These sentences would benefit from the aforementioned suggested introduction to AIW.

*__See response above – we agree__*

Li 83: Suggest to change to "increasingly, warmer, more saline, and shoaling AIW layer"

*__OK__*

Li 85 – 86: I believe that is what this paper was saying, but be sure to mention somewhere in here that this is an increase in the "overall" or "average" ice shelf melt rate.

*__OK__*

Li 89 – 90: I do not recall if Mayer et al. (2018) set out to estimate a thinning rate over the whole ice shelf. If so, state that explicitly with the thinning rate range. If not, state explicitly that this thinning rate of up to 12 m $a^{-1}$ is local only to Midgardsormen. The reader will be confused otherwise, as the Rignot and Jacobs (2002) approach suggests an increase in melting of 5 m $a^{-1}$, which can be equated to thinning if we ignore large changes in ice dynamics. This is less than half of 12 m $a^{-1}$, which is a significant difference.

*__We will reword to make clear the scope of the Mayer thinning estimate, which relates to a local thinning estimate at the Midgardsormen locality__*

Li 98: Suggest to change "then this should have profound" to a more precise statement such as, "then changes in its thermohaline properties should have profound."

*__OK__*

Li 100: Suggest to change "flux, extent, properties and interaction of AIW with the floating ice and grounding line" to "delivery of AIW to the sub-ice shelf cavity and the degree of thermodynamic interactions with the ice shelf base" to improve sentence readability.

*__OK__*

Li 102 – 104: Poorly written run-on sentence. Rewrite to improve readability.

*__OK__*

Li 106: Add the distance of Blasø from the grounding line, and reference Figure 1.

*__Blaso does not have a single distance from the grounding line (the two entrances are 10 km apart) which is why we use >50km__*

Li 107: Correct to Interferometric synthetic Aperture Radar (InSAR).

*__OK__*

Study Area:

Li 111: Is this sentence correct? Milne epishelf lake in Canada is one that comes up often in the literature that is between an ice shelf and a lodged ice mass in a fjord with fjord walls on its sides.

That is, it is not strictly bounded by an ice free land area and an ice mass. Please take a look through the literature to make sure that this statement is correct.

***We have worked on epishelf lakes for many years and this is a definition that fits both Blaso and Milne epishelf lake. We will include a slightly expanded description and context of epishelf lakes to address these comments and for the comments of reviewer #2.***

Li 111 – 112: Aren't all epishelf lakes freshwater on above the ice draft and seawater below? Isn't freshwater typically considered necessary for a body of water to be considered a lake? I suggest to correct these two sentences then combine them into one cohesive sentence that accurately defines an epishelf lake.

***We will include a slightly expanded description and context of epishelf lakes to address these comments and for the comments of reviewer #2.***

Li 114: Why must the ice be in hydrostatic equilibrium for its underside to determine the transition to seawater? Wouldn't the underside depth determine the onset of seawater regardless of the degree of flotation?

***We will include a slightly expanded description and context of epishelf lakes to address these comments and for the comments of reviewer #2. It is correct that HE is not necessary and we will correct (see reviewer #2)***

Li 115 – 119: Can you expand this thought a little further to explain concisely how epishelf lakes have been used to infer past glaciological change?

***We will include a slightly expanded description and context of epishelf lakes to address these comments and for the comments of reviewer #2.***

Li 122 – 123: Would the southern lake margins also receive freshwater from submarine melt of the ice faces and summertime runoff?

***See earlier comment that we will include a short description of the different sources/inputs of freshwater to the lake including submarine melting of the ice faces, fluvial input from the delta and other streams, supraglacial melt flowing off the ice shelf, and melting of lake ice.***

**Methods:**

Please add subsection headings to improve the organization of this section.

**OK**

Li 138 – 140: The structure of this sentence makes it unnecessarily difficult to read. Suggest to rewrite the sentence with active voice to more clearly and concisely communicate the idea.

**OK**

Li 145: Please provide a reason for the depth differences. Perhaps they are within the instrument uncertainties at these depth ranges? If so, state this uncertainty range explicitly.

***This is a measurement and spatial location issue common in any geophysical survey where line transects are being done. A crossover analysis to determine average difference in layer depth e.g. in ice-penetrating radar lines is a common statement to make. In this case it is likely due to tides, slight differences in GPS locational accuracy on sloping surfaces, and a small amount of instrument measurement error.***

Li 148 – 149: Please convert the C, T, and P uncertainties to Conservative Temperature and Absolute Salinity uncertainties, as these are the properties that the hydrographic data should be presented in.

***The CTD measures conductivity from which one can then calculate practical salinity***

*(units psu). This conversion depends on the temperature of the sample, amongst other things. Conversion to Absolute salinity will, in addition, take into account differences in the salt composition by location. Thus the sensitivity of a conductivity sensor cannot be related to an absolute salinity uncertainty in a simple way. As a result we have kept the manufacturer's specifications in the text but will explain how they are approximately proportional to values in g/kg. Similarly for Conservative temperature, thus:*

**Manufacturer-cited accuracy was ±0.01 mS/cm, ±0.01 °C, and ±0.05% for pressure. (Roughly this corresponds to an accuracy of 0.02 g/kg, and an accuracy of 0.01 °C for Conservative Temperature).**

Li 146 – 156: Were the CTD post-processed at all? If not, I suggest to post-process the data to improve their quality, as this is standard procedure in oceanography. This webpage explains nicely how to post-process the profiles: https://docs.rbr-global.com/rsktools/files/latest/57311819/57311821/1/1593023510371/PostProcessing.pdf.

**The data were processed in the ways already noted in the text. We note the MATLAB code linked by the reviewer but we are not convinced that this would add anything significant to the quality of the dataset: for example, our CTD is relatively compact and we did not think it necessary to post-process for a few cm offset between T and S sensors, and the data do not have noise that might suggest a low-pass filter would be appropriate (e.g. we have no salinity spikes at sites of strong gradients), and we sampled on a flat calm lake so loops are not present.**

Li 159: Please convert this uncertainty to a vertical range based on the observations.
*OK*
Li 162: Cite Figure 1 b at the end of this sentence.
*OK*
Li 174: Is this the annual DEM for 79 N for 2017 or is it a single DEM spanning multiple years? Please add this information to the text.
*Hydrostatic analysis used the Arctic DEM (which is constructed from data acquired over multiple years prior to 2018, and is provided at a ground resolution of 2 meters, for full details see Porter et al., 2018) for the ice surface and BedMachine v3 (which is constructed from multiple ice thickness datasets collected between 1993-2016 and is provided at a ground resolution of 150 m, for full details see Morlighem et al., 2017) for the sub-shelf bathymetry and subglacial topography*

Li 181: See above comment on correcting InSAR acronym definition.
*OK*
Li 184 – 185: I see the reference to the full InSAR processing method, but it would be nice to know in the text if tidal elevation data necessary for the vertical correction applied? If they are, did you extrapolate your measurements back in time with a harmonic analysis or use the Dansmarkhan tide gauge? Did you use the CATS2008 tide model?
*We do not correct for tidal elevation on purpose, because the intention is to identify areas affected by tides and those who are not.*

**Results:**

Please add subsection headings to improve the organization of this section.

**OK**

Li 205 – 206: Be consistent with units. Either choose cm or m for tidal range. Figure 2 presents amplitude as cm, so perhaps that is the best route forward.
***Fair comment – we will be consistent***

Li 208 – 211: I do not understand the point of this text. Is it just saying that the CTD profiles were retrieved within some range of the seafloor? I think that most of this is dead text that can be cut out so that this paragraph can be combined with the next to more concisely state the results.
***This is pointing out that the CTDs necessarily sample quite different (maximum) depths due to the locations they were taken. We do not think it is dead text and is a necessary part of explaining the context behind the CTD profiles in Fig 3.***

Li 212: Replace "haloclines and accompanying thermoclines" with "pycnoclines where temperature and salinity increased rapidly with depth."
**We will use pycnocline and add a statement that the density changes are likely driven dominantly by Salinity, thus: "with pycnoclines associated with changes in salinity given how dominant salinity is in governing density at these cold temperatures"**

Li 225 – 226: Is this sentence referring to the upper limit of the pycnocline or its thickness? This is not clearly communicated with this sentence.
***It is a difference in the measured depth of the boundary – we will clarify.***

Li 227 – 229: This is a very interesting finding that in its current form is kind of a distraction from the narrative. Please add a temperature and salinity range where these fish have been observed to live in to fix this. Also, this is minor, but were the fish dying or dead? If they were interpreted as dying, what was the reasoning for this interpretation. Please write more precisely.
***We internally debated whether to include this but we think that they add an important implication that marine water must be reaching the western basin from under the ice shelf. We saw both dying (gaping, flapping, and unnaturally floating on their sides 'high' in the water) and dead fish. We know of no source for the T-S range but as with all cod, they are normally found in fully marine conditions and have not been reported in lakes. The Froese and Pauly reference is to FishBase which we understand to be the leading marine fish database. We will add some wording to clarify.***

Li 233 – 235: Poorly written run-on sentence that contains multiple thoughts. Please rewrite as two sentences.
***OK***

Li 243: Correct to "free floating (Fig. 5d)."
***We will add the space***

Li 263: Suggest to change to "ice penetrating radar data," because there are many different frequencies of radar used to measure ice and some do not penetrate through the whole ice column.
***OK***

Li 285: Please convert these data ranges to months so that the reader can more easily compare the lags presented in this manuscript to those in Wilson and Straneo (2015) and Schaffer et al. (2020).
***OK***

**Discussion:**

Li 273: Please change "melt-mixing line" to "meltwater mixing line (Gade, 1979)." I've added the citation to the reference section below.
***Ok***

Li 273: Since this is the first mention of glacial modification of AIW through melt input, I suggest

to clarify this as "glacial melt modified AIW" or "glacially modified AIW" and to clarify that this is colder and fresher (less thermal driving) than pure AIW.

*OK*

Li 274: Again, the ITM was deployed through a rift, but all the sensors resided beneath the ice shelf base, so I think "deployed in the rift" is slightly misleading.

***See earlier response.***

Li 276: The expression "50 km inboard of the calving front" is somewhat awkward. Consider using a different word such as "upstream, upglacier, westward" or something similar.

*OK*

Li 283: The reference to the Schaffer et al. (2020) paper is awkwardly placed. I suggest to place it after "grounding line."

*OK*

Li 296 – 299: Poorly written 59 word run-on sentence containing at least two ideas. Please rewrite this sentence and break it into two shorter sentences.

**Will split in 2.**

Li 300: I suggest to use "pycnocline" instead of "halocline", because both temperature and salinity characteristics are referred to here.

**We will use pycnocline and add a statement that the density changes are likely driven dominantly by Salinity, thus: "with pycnoclines associated with changes in salinity given how dominant salinity is in governing density at these cold temperatures"**

Li 302 – 303: Why does the 145 m pycnocline have to be a proxy for the ice shelf draft "in hydrostatic equilibrium?" Why can't it simply reflect the ice shelf base depth?

***Covered by earlier response on epishelf lakes.***

Li 300 – 305: There is another fresh water mass that is not considered here that will fill the lake. That is glacial meltwater from submarine melting. Please update this paragraph to reflect this.

***Covered by earlier response on freshwater sources.***

Li 306 – 309: I do not think that this is the proper mechanism to explain the 5 m difference in pycnocline depth considering the insane vertical stratification across this feature. Internal wave amplitudes decrease with stratification and their speeds increase, so I would expect them to be much smaller and quite fast. I would expect that it is more likely that the seawater layer simply goes up and down with the tide, then there is quite a strong pressure driven flow into and out of the lake where water parcels flow down isopycnals. Please expand the discussion to reflect this comment or include a rebuttal, with a scaling argument in the text, that defends the internal wave interpretation.

**We note that the internal wave explanation was picked up by this reviewer and by the third set of comments. We agree with the reviewer that we do not have enough information to fully understand the 5m difference in pycnocline depth observed between CTD5 and CTD8. Since this difference does not affect any of our results - we have modified the text to avoid attribution and we will include the potential explanations of errors introduced by tides and instruments, internal waves, and add the mechanism noted in the third set of comments and will summarise that we do not have an unequivocal explanation for the difference in depth of the boundary between the two sites. This does not affect our overall conclusions.**

Li 322: Is the tidal exchange unencumbered? Earlier in the text it was hypothesized that the somewhat reduced tidal amplitudes results from the partially grounded margin of 79N Glacier inhibiting tides. Please clarify this.

***We will remove 'unencumbered'***

Li 337: Please change to "dying marine fish" so that the reader knows at this point, not later, that Arctic Cod are strictly marine. Also, see prior comment above that points out that it would be helpful to state the thermohaline range that these fish live in.

*See earlier response.*

Li 3338: The statement that there is "no seawater present in the deepest parts of the western basin" is incorrect, because brackish water contains seawater. Please correct this portion of the text to be more precise.

*We will change to 'there is no fully marine water'*

Li 344: Ok, so this is way outside of my area of study, but wouldn't fish that live in denser seawater and therefore balance their buoyancy to the denser water, sink if placed into less dense brackish water?

*Also outside our primary expertise. But from biological texts it seems that osmotic lysis is the simplest explanation, whereby the change in osmotic potential of the cells of a saltwater fish that finds itself in a hypotonic solution would be for the cells to absorb freshwater, bloat and rupture. Whatever the mechanism it is clear that we found marine fish floating in distress in fresh surface waters. We will clarify that we were not referring to swim bladder buoyancy adjustments but to cell osmosis.*

Li 350 – 352: Ok, I do not disagree that 79N Glacier appears to be thinning significantly, but how do several CTD profiles from a single year that reveal seawater in the lake show this? They cannot resolve change because they are from a single point in time – they just show that there is seawater at depth in the lake. The migrating margin I believe shows this, but this is not explained sufficiently here, and I don't think these data are really shown in this manuscript. Please expand this discussion to more sufficiently defend the reasoning that Lake Blasø can tell us about the thinning 79N Glacier. I believe that this will require at least an additional paragraph, perhaps two.

*We will clarify that the conclusion we make here is in the context of some of the other work such as that by Mayer et al, where we can combine their conclusions with our observations to suggest that the incursion of marine waters \*may\* be relatively recent.*

**Conclusions:**

Li 359 – 360: It is important to clarify that the AIW at the ITM site that is being referenced has already been glacially modified. At least this is what I thought was communicated earlier in the text. Either way, it would be helpful to clearly communicate the temperature and salinity difference between the AIW observed in Lake Blasø and the warmest and most saline AIW observed at the ice shelf front. The Schaffer et al. (2020) or Lindeman et al. (2020) reference should provide the necessary information.

*See earlier response. We will quantity the different in T-S ranges of AIW between ITM and Blaso*

Li 373 – 377: This statement ignores the logistical difficulties of establishing a continuously-monitoring mooring in the western basin of Lake Blasø, which will be significant. Calving icebergs from the tidewater fronts will likely have a draft close to the maximum depth of the fjord and will have a high likelihood of running into the mooring. If this statement is to be left in the text, then there should be a disclaimer about the inherent risk in establishing a mooring in the western basin.

*We will remove 'continuous' – the innovative suggestion we are trying to make is that this is logistically a much easier place to deploy a (campaign) mooring than drilling through an ice shelf.*

Li 881: Suggest to change to "glacial meltwater from basal melting of the ice shelf."

*Line 381 – OK.*

**Figures:**

Figure 1: Consider adding a reference map that places NEGIS in context of Greenland as a whole, and adding glacier demarcations to the inset figure. I believe these data can be downloaded here http://imbie.org/imbie-3/drainage-basins/. Add a northward-pointing arrow or longitude and latitude lines to panel A. It would be nice to have several more ticks on the color bar to more easily identify lake depths in relation to the color scale.

***We will improve Fig 1 for visualization of the bathymetry and will add a map of ice shelf thickness and a long profile of the fjord (See also Reviewer #2)***

Figure 2: Perhaps it would be clearer to present these data as deviations about a 0 cm elevation. That would make the data easier to compare and would represent tidal fluctuations more accurately.

***We will follow the suggestion of Reviewer #2 to align the observations more clearly.***

Figure 3: Please label x axes as Conservative Temperature (Θ) and Absolute Salinity ($S_A$)

**OK**

Figure 4: Please label axes as Conservative Temperature (Θ) and Absolute Salinity ($S_A$). Also, The caption says that the blue data are from the Eastern Basin, but the legend labels them as the Western basin. Please correct this.

***OK – we will correct the legend which has east and west transposed.***

Figure 5: This is a very informative figure, but in its present form it takes up 2.5 pages and requires some improvement. I suggest the following changes:
- I suggest to either split into three figures or remake what is currently Figure 5 so that it all fits on one page.
- Additionally, the vertical scales on d – f) are different. It would improve the interpretation of this figure if these panels all had the same vertical scale or the panels heights varied with respect to the vertical scale.
- The inclusion of a filled space beneath the ice shelf that is either modified AIW or fjord water is misleading. The authors do not data to prove this. Please either label this part of the figure as AIW (interpreted) or something similar. In reality the sub-ice shelf water column will have up to 4 different water masses mixed into it, with AIW probably being the dominant water mass.
- I suggest to interpolate between multiple CTD profiles to fill in the Lake Blaso water column structure.
- Finally, I suggest to replace the multiple legends with a single legend and nest the maps of the lake locations in the empty space to improve the figure.

***We will consolidate this figure to occupy less space and to pick up the errors picked up by this reviewer and Reviewer #2.***

**References:**

Wilson, N., Straneo, F., & Heimbach, P. (2017). Satellite-derived submarine melt rates and mass balance (2011–2015) for Greenland's largest remaining ice tongues. *The Cryosphere*, *11*(6), 2773-2782.

Gade, H. G. (1979). Melting of ice in sea water: A primitive model with application to the Antarctic ice shelf and icebergs. *Journal of Physical Oceanography*, *9*(1), 189-198.

---

## Author Comment (AC2)

I apologize to the authors and journal editors for being late with this review. I got carried away with other responsibilities but thoroughly enjoyed reading the manuscript and appreciate the opportunity to comment on it.

This manuscript blends geophysical, remote sensing, hydrographic, glaciological, oceanographic and limnological datasets together to 1) demonstrate conclusively that Blåsø, a fresh/brackish body of water at the ice shelf margin, is an epishelf lake and to 2) argue that Atlantic Intermediate Water (AIW) can reach the grounding line 79N glacier and is interacting with the ice there. The paper is well presented and of significance to the readership of this journal. I have no hesitation recommending it be accepted for publication in The Cryosphere as long as some minor comments/suggestions are addressed.

Derek Mueller, Carleton University [2022-12-17]
***Many thanks for the review and for the positive comments. We address the suggestions and comments individually below.***

**General comments**

*Epishelf lake:*
To my knowledge this paper is the first to describe Blåsø as an epishelf lake. These lakes are rare and unique so it was somewhat surprising that the significance of this was not highlighted as much as it could have been.

Gibson and Anderson (2002) was cited and it might be a good idea to explain examine Blåsø within the framework they illustrated in their Figure 2 where there are two types of epishelf lakes – Type 1, "with freshwater directly overlying marine water" (I assume this is the case for the east basin) and Type II – "with indirect connection to the marine environment" (perhaps a more suitable description for the west basin – if there is a conduit on that side?). There could also have been more description of Blåsø. How big is the catchment? How much of it is glacierized? How common is summer ice cover? Is there more to say about water mass 2 in the eastern basin and water mass 3 in the western basin? How did they form and how does they persist?

You write [for a Type 1 epishelf lake] "the depth of the transition between marine and brackish/fresh water is controlled by the draught of the floating ice" but this should really be the *minimum* draft [draught] of the ice shelf (whether this point is in local hydrostatic equilibrium or not). A caveat here is that epishelf lakes can over-deepen in the summer due to freshwater input (see Hamilton et al., 2017 and Bonneau et al. 2021).

As as consequence of the above, it is challenging to find the minimum draft of the ice shelf that is controlling the outflow. The radar transect presented in Fig 7 is likely the best approach available, far better than estimating draft using hydrostatic equilibrium. So, I agree that the airborne radar (and InSAR) are more reliable (stated on ms line 328), although it is fine to include all the data for context. The fact that the minimum draft you highlight (150 m) is so close to the interface between water mass 2 and 4 is pretty convincing (but also see below).
***We will add some extra description and context for epsihelf lakes and for Blaso and its***

*catchment to address these comments and those from Reviewer #1. The suggestion to more effectively use Gibson and Anderson as a framework is very helpful.*

*Uncertainty/Errors:*
It would be helpful to know more about the uncertainty of the various datasets that are used in the analysis as most of the data is presented without any associated error bars.  The propagation of errors and comparisons of the datasets can then be discussed.   For example:

- The water mass 2 - 4 interface varied by 5 m (n=2) so how well is this constrained?

*See response to reviewer #1 – we will expand this discussion but the difference is well outside measurement uncertainty.*

How are bedmachine data produced?  What error is associated with this product under grounded and floating ice?
- Is it reasonable to follow Morlighem et al., (2017) and assume an ice density of 917 kgm-3 and sub-shelf water density of 1023 kgm -3?  You may not have any ice density measurements on hand (not many do), but is this water density realistic, given data from Figure 4, water mass 4?
- What about DEM precision and accuracy?  How does this and density uncertainty impact the HE calculations?

*Bedmachine v3 is currently state-of-the-art for the subglacial topography for Greenland (and the same applies to the ArcticDEM for the ice surface). The production of both have detailed processing workflows but an explanation of these is beyond the scope of this paper but we have provided brief additional clarification and we have made direct reference to the source publications. In relation to the choice of the ice shelf and sub-shelf water densities: we used the values applied by Schaffer et al., 2016 (not Morlighem et al., 2017 as we originally stated – this has now been amended). We chose the lower value for water sea density in order to be conservative in our calculations i.e. while we have indication of a direct connection in the eastern basin we do not have the same in the western. However, we recalculated the ice shelf draft assuming a water density of 1027 kgm-3, which resulted in only a minor change to the grounded ice in the west and no change in the east (see screen grabs). So the choice of densities within this range has no influence on our results.*

- What is the error in the radar thickness/draft and InSAR?

*The geographic precision of the flight trajectory and so the radar and laserscanner data is around 0.05m. Uncertainty of the GNSS altitude and thus laserscanner elevation is usually within 0.1m. The laserscanner is calibrated using runway passes and runway crossing. The uncertainty of ice thickness is usually within 20m.*

*Oceanography and ice-ocean interactions:*
I have not read Lindeman et al. (2020) but I feel there are probably answers to the following questions within it.  These could be brought into the text so that it is clear in your manuscript without readers having to go elsewhere to find info.

*See response to Reviewer #1. We will include a longer description of AIW and of the Ice-tethered mooring site.*

There is mention of AIW being present at 500 m in the rift mooring data but I have the sense from the description that this varies over time.  Please explain the dynamic nature of the AIW – what depths is it found at, what range in salinity and temperature, what is above (and below, if

relevant) this water mass?.  It would be helpful if this explanation also cleared up why you match Blåsø water properties to the ITM data at a specific moment in time.  How consistent is this water at the rift ITM?

*See above.*

Figure 4 shows mixing lines from the 500 m level at the ITM mooring at 3 times of year and some daily values (unclear what time of year they are from).  These melt lines align with water mass 4 properties and this is used to infer that AIW interacted with the ice shelf at some depth (presumably between 500 m and ~200 m).  But are there any other water masses or combinations of water masses that could account for the water properties seen year?  In other words, are there any alternative hypotheses to explore before espousing your interpretation that AIW is the culprit?

**We will clarify the timing by noting labels in panel a apply also to b and by making the explicit links between the ITM oceanographic measurements and the modified-AIW at depth in Blaso.**

To tie together the above 2 paragraphs, if you had a profile in the east basin from January 2018 would you hypothesize that it would be on the July 2017 melt line?

**See above**

Lastly, I think adding bathymetric contours to the map in Figure 1 or elsewhere and an along fjord cross-section of the bathymetry, and ice draft and elevation to complement text on lines 45-50 and 60- 70 would be very helpful.

*We will include an along-fjord profile in Fig 1 and a map of ice shelf thickness. See other responses on bathymetric contours.*

**Detailed minor comments**

line 35 - exhibited little response to atmospheric and oceanic warming [in or over] the decades

*OK*

line 38 - Model projections suggest that ocean warming around Greenland will double.  Do you mean the rate will double or the temperature relative to some reference period?  Explain

*We will correct – this should read 'will be double the global mean ocean warming by 2100'.*

line 47 - grounding line (~600 m below sea-level).  This doesn't seem to match the ice shelf thickness of 300 to 100 m.

*Inclusion of the along-fjord profile will help clarify this sentence.*

line 88 – Midgardsormen ridge description is unclear here.  Why did it flow backward?  How much landward migration was there.  This feature becomes clearer later in the text but it would help to have a better explanation here.

*We will attempt to clarify the configuration of Midgardsormen by bringing some text forward, but we are unsure why the reviewer  refers to 'backward' flow. We will revisit to clarify text.*

line 104-108 – You could include characterizing Blåsø as a specific purpose of your paper in the statements here

*OK*

line 107 - synthetic Aperture Radar Interferometry  (capital S)

*OK*

line 113 - The depth of the transition between marine and brackish/fresh water is controlled by the [minimum] draught of the floating ice.  This is true if the adjacent ice shelf is not grounded (whether or not it is perfectly in HE or not).

*Yes – see response to Reviewer #1 where we have agreed to add some extra text on epishelf lakes.*

line 143 - CHIRP (Compressed High Intensity Radar Pulse) - I am really unclear on this. Does radar actually work underwater? Are you not using sonar?

**It is odd – although it a sonar system the manufacturers still use CHIRP as a descriptor, presumably inherited from nomenclature of radar systems because they use the same principle of sampling multiple frequencies as the original CHIRP radars. We will add the word 'sonar' to read 'CHIRP sonar', and thus remove ambiguity.**

line 148 – For pressure is that +/- 0.05% of the pressure value or the full scale – if the latter, please share what that is.

*We will clarify by giving this as an equivalent water depth measurement.*

line 159 – please give full scale or convert to accuracy in pressure units

*And for this one too.*

line 225 34.4 to 34.7 g/kg. change to gkg-1 to be consistent

*See response to reviewer #1 – we will make these (and other) units consistent*

line 276 – 79N Glacier [capital G]

*See response to reviewer #1 – we will use consistent nomenclature*

line 315 – thank you for explaining the 5 m discrepancy – It is true there are internal waves in Milne Fiord epishelf lake. I certainly don't recall them being on the order of 5 m. Maybe there is another explanation for this?

*See response to Reviewer #1 – it is helpful to know the magnitude of the Milne fjord internal waves. This discussion will be expanded but the emphasis on internal waves reduced.*

line 317 79N Ice Shelf – proper name capitalization.

*See above*

line 357 – replace measurement x1 with a synonym to avoid redundant text

*OK*

line 363 – can you give error/uncertainty here?

*Not really – these are complex spatial patterns and don't translate well into simple quantifiable differences. We think they are well illustrated in Figs 5-7.*

Figure 1A
Would it be possible to see a bit further to the west?
Would it be possible to add some bathymetric contours to this
figure? Why are there 2 ESA CCI 2017 grounding lines?

*See response to reviewer #1 – We will improve Fig 1 for visualization of the bathymetry and will add a map of ice shelf thickness and a long profile of the fjord (See also Reviewer #2)*

Figure 1B
Would it be possible to krig and contour the bathymetry in addition to the CHIRP data? The colour ramp could use some intermediate values between 0 and 212 m.

*Yes on the colour ramp, but kriging would not be supported for our spatial density of sampling.*

Fig 1 caption
Red dots show where moorings have measured [water] flow direction [no s – only one arrow] in (yellow arrow)
SG = Storstrømmen Glacier – remove comma

*OK*
 Figure 2 – the grey line is hard to see.
 How far away is the Dansmarkhan tide gauge?
*OK*

 Each of the 3 records could be centered on the tide gauge data by offsetting by the difference
 in the average values of coincident records.  They would still have arbitrary datums but
 would be aligned.
***Thanks – nice idea and we will do this.***
 Figure 4a  caption
 The shaded range indicates the instrument accuracy  - do you mean precision?
 Daily values -from ITM – are these for specific times of year or for the entire record?
***We will correct to 'precision'.***
 Figure 5a
 Unclear what the red dashed line is
 The dashed grounding line are 2 mutually exclusive options for how the GL could go?  It
 should be clear.
 Midgarsormen line is green in the figure and yellow in the
 legend. Need a legend for the toppography (colours)
 Note the inset maps c and d might be better placed in the cross sections under 5d, e and f.
 (along with the transect).  There should be space for them there under the lake – with their
 own scale bar too.  The legends can move over to the right

 Fig 5e – there is no partially grounded dashed
 purple line Fig 5f – explain the 6c an 6b lines

 Figure 5 caption -
 Midgardsormen (yellow line)  is green
 eastern calving front at the point where the extent of grounded ice is narrowest (see Fig 5).  - but
 this
 _is_ Figure  5 – do you mean somewhere specifically?
***See response to reviewer #1. We will consolidate this figure to occupy less space and to pick
up the errors picked up by this reviewer and Reviewer #1.***

 Figure 6
 In E, short arrows show possible grounding of the Midgardsormen.  There is no E
***We will correct to (b)***
 Figure 7
 The numbers in b are very small and hard to read
***We will expand***
 Table 1 – need to add degree symbol and minute symbol
*OK*

**Citations I made that are not already in the manuscript**
Bonneau, J., Laval, B. E., Mueller, D., Hamilton, A. K., Friedrichs, A. M., and Forrest, A. L.:
Winter dynamics in an epishelf lake: Quantitative mixing estimates and ice shelf basal channel
considerations,

J. Geophys. Res. Oceans, 126, e2021JC017324, https://doi.org/10.1029/2021JC017324, 2021.

**Citation in our response**

Schaffer, J., Timmermann, R., Arndt, J. E., Kristensen, S. S., Mayer, C., Morlighem, M., and Steinhage, D.: A global, high-resolution data set of ice sheet topography, cavity geometry, and ocean bathymetry, Earth Syst. Sci. Data, 8, 543–557, https://doi.org/10.5194/essd-8-543-2016, 2016.

---

## Author Comment (AC3)

**Comments on *Direct measurement of warm Atlantic Intermediate water close to the grounding line of Nioghalvfjerdsfjorden (79N) Glacier, North-east Greenland**

**Global statement**

Interesting and timely results of Atlantic water close to the grounding line of 79N. I think this manuscript should be accepted after the comments of the two reviewers are addressed. I also have some minor comments that I believe would improve the manuscript.

*Many thanks for taking the time to look at this manuscript, and for your positive introductory comments. We address your specific comments and suggestions below.*

**General comments**

Neither An et al .(2021), Lindeman et al. (2020), Mayer et al. (2018) or von Albedyl et al. (2021) present a full map of the ice shelf (glacier tongue) thickness. I feel since you are already using ArcticDEM for areas near the epishelf lake, you could present an ice thickness contour map of the ice shelf. That would not be too much work if you already have the grounded ArcticDEM data. Maybe that would help explain the weird circulation pattern outflowing through Dijmphna Sund.
*We will add a map of ice shelf thickness to Fig 1.*

I did not see any discussion about subglacial discharge. I think it is warranted to at least briefly discuss why it is not relevant in this study. You are really close to the grounding line and your CTDs are taken in summer, this is prime location and timing to see subglacial discharge. You mention that a "simple plume model would account for the thinning […]". Subglacial discharge is sometimes included in these plume models, maybe that's a place to emphasize subglacial discharge does not matter so much in this case.
*See response to reviewer #1 where we note we will add an explanation of the different freshwater inputs to the lake*

I agree with reviewer 1 comment that using "water mass" is misleading.

I would add a few sentences on why "water mass 2" in the eastern basin is fresher than "water mass 3" in the Western basin. Isn't the eastern basin more connected to the ocean than the western basin?
*See response to Reviewer 1 on the 'layers' or 'masses' point.*

**Edits**
L36: "Unlike many other sectors of the Greenland Ice Sheet, NEGIS and the ice shelves that front it exhibited little response to atmospheric and oceanic warming for the decades immediately prior to the mid 2000s." Maybe add the citation(s) to this statement directly at the end here instead of after the next sentence.
*OK*
L46: "NEGIS flows at ~1200 m a-1 and upstream from the 79N Glacier grounding line (~600 m below sea-level) the basin floor deepens to ~1000 mbsl (Bamber et al., 2013)". Is that velocity at the grounding line of 79N? Maybe divide this sentence in two, one about the velocity (with where the

1200 m a-1 velocity is taken) and one about the bathymetry.

*OK*

L56: "Humbert et al (submitted) also identify a recent shift in calving style and fracturing at the calving front of 79N Glacier." A shift from which style to which style? Maybe also add a sentence on the implications of this statement.

*OK*

L59: "Based on the rapid decrease in thickness of the ice shelf, to only 330 m within 5 km of the 79N Glacier grounding line ,…"  To 330 m, ok, but from what thickness?

***Will be clarified by inclusion of along fjord profile in Fig 1***

L69: "There is only one measurement of AIW in the cavity, where it has been detected in a rift in the 79N ice tongue, located ~10 km behind the calving front" When was this measurement? Is this from the ITP? If so make this clearer.

**OK**

L96: "groundling" [typo]

**OK**

L98: "If AIW is circulating throughout the cavity beneath the floating portion of the 79N Glacier then this should have profound consequences for stability of the grounding line (An et al., 2021) and the ice shelf." Change "floating portion" by ice shelf or glacier tongue and be consistent throughout the paper, i.e. no alternating between both.

***See response to reviewer #1 on use of ice shelf.***

L112: "Where a source of freshwater feeds into the lake [add coma] a salinity-driven stratification forms with the more saline marine layer capped by a freshwater layer."

**OK.**

 L138: "As part of a wider programme to characterise and sample water and sediments in Blåsø, to understand past changes in the 79N Glacier, the bathymetry of the lake was mapped and multiple CTD profiles were measured in different parts of the lake." Sluggish, maybe break into two sentences?

 **OK**

L146: While at it, what's the sampling rate of the CTD and how fast was it lowered? How was the data averaged/binned?

***We will note these details.***

L175: Partially out of personal interest, but likely relevant to many others: did you compare bedmachine3 to your CHIRP survey? If so, one or two sentences on the comparison would definitely be relevant.

***Only where they intersect in Fig 5. The comparison of Fig 5 with the radar profiles can show that – at least close to the eastern entrance to Blaso - Bedmachine is not correct.***

L208-211: Use either just CTD or just CTD profile or water profiles, but be consistent.

*OK*

L227: "During bathymetric and CTD surveys [add coma] we observed …"

*OK*

L308: "It is more likely that the difference is caused by internal waves which can be created where tidal currents drive water parcels, especially, on steep slopes (Munk and Warren, 1981)". You mean baroclinic tides? Any evidence of baroclinic tides in the ITM or other moorings  record? This is a pretty big density jump, 5 m internal waves would be quite impressive. Could it be just that what you call water mass 2 is draining away? (making its way out through cracks and channel). Seasonal cycles of deepening and shoaling have been reported in Milne Fiord epishelf lake (Hamilton 2017, Bonneau 2021).

***See response to Reviewers #1 and #2 where we note we will include an expanded discussion on the potential explanations for the difference between CTD5 and 8. Thanks for the references.***

L316: Again, possible internal waves, but I would not conclude it is without a doubt based on two CTD profiles. High frequency internal waves in Milne Fiord epishelf lake have a maximum amplitude of 15 cm and a period of 50 min.

*See above.*

Figure 1A: Would it be possible to add some bathy contours from bedmachine3? Need a scale for the velocity arrows.

*We think this is beyond the dataset but will include an along-fjord profile*

Figure 1B: I think you have enough data to generate a decent bathy map of the lake. It would be nice see that. And overlay the data points?

*We do not think that we have sufficient spatial sampling to do this robustly and it risks generating spurious bathymetric features, highly dependent on kriging/interpolation methodology.*

Figure 3, 4: You say you are using conservative temperature and absolute salinity. These should be your labels.

*OK*

Figure 4: Are the dots your CTD data points and the lines just link the data points? If so I would remove the lines and make the markers a little larger, as it is usually done on T/S diagram.

*We will expand symbol size but we will keep the lines: they are helpful to show the depth relationship visually in T-S space and to separate the different profiles.*

Figure 5D: Check y axis, something is wrong.

*We will correct -100 and -200 m depth markers.*

Table 1: Not sure it brings something to the paper, could easily do without.

*We think these are important to show where the CTDs were taken.*